# Modeling Dragons: Using linked mechanistic physiological and microclimate models to explore environmental, physiological, and morphological constraints on the early evolution of dinosaurs

David M. Lovelace[1]*, Scott A. Hartman[2], Paul D. Mathewson[3], Benjamin J. Linzmeier[2], Warren P. Porter[3]*

1 University of Wisconsin Geology Museum, Department of Geosciences, University of Wisconsin-Madison, Madison, Wisconsin, United States of America, 2 Department of Geoscience, University of Wisconsin-Madison, Madison, Wisconsin, United States of America, 3 Department of Integrative Biology, University of Wisconsin-Madison, Madison, Wisconsin, United States of America

* dlovelace@wisc.edu (DL); wpporter@wisc.edu (WP)

**Data Availability Statement:** All relevant data such as experimental parameters and preliminary results

## Abstract

We employed the widely-tested biophysiological modeling software, Niche Mapper™ to investigate the metabolic function of the Late Triassic dinosaurs *Plateosaurus* and *Coelophysis* during global greenhouse conditions. We tested a variety of assumptions about resting metabolic rate, each evaluated within six microclimate models that bound paleoenvironmental conditions at 12° N paleolatitude, as determined by sedimentological and isotopic proxies for climate within the Chinle Formation of the southwestern United States. Sensitivity testing of metabolic variables and simulated "metabolic chamber" analyses support elevated "ratite-like" metabolic rates and intermediate "monotreme-like" core temperature ranges in these species of early saurischian dinosaur. Our results suggest small theropods may have needed partial to full epidermal insulation in temperate environments, while fully grown prosauropods would have likely been heat stressed in open, hot environments and should have been restricted to cooler microclimates such as dense forests or higher latitudes and elevations. This is in agreement with the Late Triassic fossil record and may have contributed to the latitudinal gap in the Triassic prosauropod record.

## Introduction

Paleontologists have long inferred the biology of extinct organisms from morphological correlates and paleoenvironmental context. Hypotheses derived from morphology rely on extant phylogenetic bracketing and modern analogs for support [1,2]; lack of inferential specificity may come from trimmed phylogenetic trees or increased distance from extant relatives [3]. Spatiotemporally derived hypotheses suffer from confounding factors related to bias in the

(and their parameters) that are not listed in the manuscript and its Supporting Information files are reposited with the Data Archiving and Networked Services (DANS): https://doi.org/10.17026/dans-238-9pjs.

**Funding:** The author(s) received no specific funding for this work.

**Competing interests:** The authors have declared that no competing interests exist.

stratigraphic record [4–6]. These biases can be tempered by using physics-based constraints to model a broad range of paleobiological phenomena.

All animals, living and extinct are constrained by physics. Gravity exerts control on the maximum size attained by terrestrial clades, from spiders to sauropods [7], and biological thermodynamics constrains the rate heat is produced as well as its transfer to and from the environment [8]. Robust biophysiological models, such as Niche Mapper™ are built on the fundamental principles of heat and mass flux into and out of individuals [9]. Morphology (e.g. posture, insulation, color, and body part dimensions), behavior (e.g. seeking shade, sunning, fur or feather erection, varying activity level and location), and physiology (e.g. metabolic rate, peripheral blood flow, respiratory and cutaneous water loss) can accelerate or retard heat exchange with the surrounding environment, setting temporal and spatial constraints (boundary conditions) for animal function.

For decades ecologists have modeled the physics of heat transfer to understand ecological and biogeographic constraints of modern organisms [10–25]. Biophysiological models have only been sparsely applied to deep time investigations [e.g. 26,27]. For paleoecologists, the paleobiogeographic distribution of an extinct organism is harder to test with respect to organismal physiology. For instance, it has been noted that there is an absence of large (>~1000 kg) prosauropod dinosaurs in the well-studied tropical to subtropical latitudes during the Late Triassic (e.g., the Chinle Formation of southwestern U.S.), while smaller (<~100 kg) theropod dinosaurs and their closest relatives are quite common [28]. In contrast, both large and small (~10–100 kg) prosauropods as well as small theropods are well represented in temperate latitudes. A number of hypotheses have been proposed to explain this pattern, including exclusion via high temperatures and environmental fluctuations, increased severity and frequency of wildfires, aridity, and precipitation patterns [28]. These hypotheses are difficult to test explicitly without the use of biophysical modeling.

The paleobiogeographic distribution of *Plateosaurus* and *Coelophysis* along with associated paleoclimate proxies from fossil localities provide boundary conditions to test hypotheses of environmental exclusion due to physiological limitations. We applied Niche Mapper to test a range of physiological possibilities for two adult-sized Late Triassic dinosaurs, the ~20 kg theropod *Coelophysis bauri* (Cope 1887) and the ~1000 kg prosauropod *Plateosaurus engelhardti* (von Meyer 1837) to quantitatively test the thermal constraint hypothesis. We performed sensitivity analyses to determine the relative contributions of climate, body mass, shape, diet, insulation, and the efficiencies of respiratory, digestive, and muscle systems, as well as feasible daily core temperature range and resting metabolic rate on total energy requirements. The integration of biophysiological and microclimate models offers a potentially powerful means of testing feasible physiologies and behaviors of extinct organisms against known fossil distribution and their associated paleoenvironments.

## Materials and methods

We employed the mechanistic modeling program Niche Mapper, developed at UW-Madison [29]. Niche Mapper is compartmentalized into generic microclimate and animal submodels that each contain momentum, heat, and mass transfer equations. The microclimate model has 51 variables relating to seasonality, insolation, shade, wind, air temperature, humidity, cloud, and soil properties. The biophysical model is composed of 270 morphological, physiological, and behavioral parameters previously described in detail and tested over a wide range of animal taxa including reptiles, amphibians, birds, mammals, and insects [e.g., 12,13,15,16,23,30–39]. The user is able to control how many days (1 to 365) and how those days are distributed

throughout the year, and how many of the variables (such as air temperature, feather density, or body mass) vary for each modeled day (see S1 Appendix).

Fundamentally, Niche Mapper calculates hourly energy and mass expenditures that can predict survivorship based on reasonable bounds of thermal stress and resource availability. Paleoenvironmental bounds for the microclimate model are derived from environmental proxies preserved within the rock record and data from analogous modern environments, when applicable. Measurements of skeletal dimensions parameterize a simple geometric volume that approximates the shape of the animal. The range of metabolic rates known from modern tetrapods bounds modeled rates. These boundaries allowed us to explore a reasonable parameter space although the model could easily be extended to explore unique circumstances and test novel hypotheses.

As an additional test of our Triassic microclimate models and as a point of comparison we modelled *Varanus komodoensis* (Komodo dragon), using the same methodology implemented for our dinosaur models. We compared our modeled results against observational data for *V. komodoensis* [40–42] activity patterns, food requirements, body size and body temperature (S2 Appendix).

## The microclimate model

The microclimate model calculates hourly air temperatures, wind speeds, and relative humidity profiles, solar and thermal long wavelength infrared radiation, ground surface and subsurface temperatures available to an animal [11,29,43,44]. Local environments at mean animal height are used for heat balance calculations (Fig 1).

The microclimate model fits a sinusoidal curve to user-specified maximum and minimum daily air temperatures, wind speeds, cloud cover, and relative humidity to estimate hourly values. We set minimum air temperature, minimum wind speed, maximum relative humidity and cloud cover to occur at sunrise. Maximum air temperature, maximum wind speed, and minimum relative humidity and cloud cover are set to occur one hour after solar noon [45]. Clear sky solar radiation is calculated based on date, hour of day, and geographic location adjusted for cloud cover and overhead vegetation [43,46]. Paleosolar calculations for insolation were computed from Laskar et al. [47] using the *palinsol* program in R [48]. Because of uncertainties in deep time insolation we used modern values as a first step. However, we recognize there is variability in insolation not only in deep time, but also on shorter timescales due to precession, obliquity, and eccentricity. Cloud cover also reduces solar radiation intensity at ground level and provides thermal cover by trapping longwave radiation that would otherwise escape to the sky, increasing the sky's radiant temperature [29,44,45,49].

Long wavelength thermal radiation from clear sky and clouds were computed using empirical air temperature correlations from the literature [50,51]. Substrate thermal radiation was computed hourly from the numerical solution of a one-dimensional finite difference transient heat balance equation for the ground. Hourly outputs from the microclimate model specify above and belowground local microclimates in the sunniest and shadiest sites specified by the user.

**Microclimate model parameterization in deep time.** In order to test the hypothesis that large dinosaurs such as *Plateosaurus* were restricted from tropical latitudes primarily due to thermal constraints we chose to use the well sampled Late Triassic strata of the Chinle Formation of western North America as a model for our Late Triassic paleoenvironment. Although these strata are currently located at 35 degrees north, the Chinle was originally deposited between 5 and 10 degrees north paleolatitude [52]. We used published local and regional paleoclimate data derived from the Chinle Formation; these data include sedimentary proxies

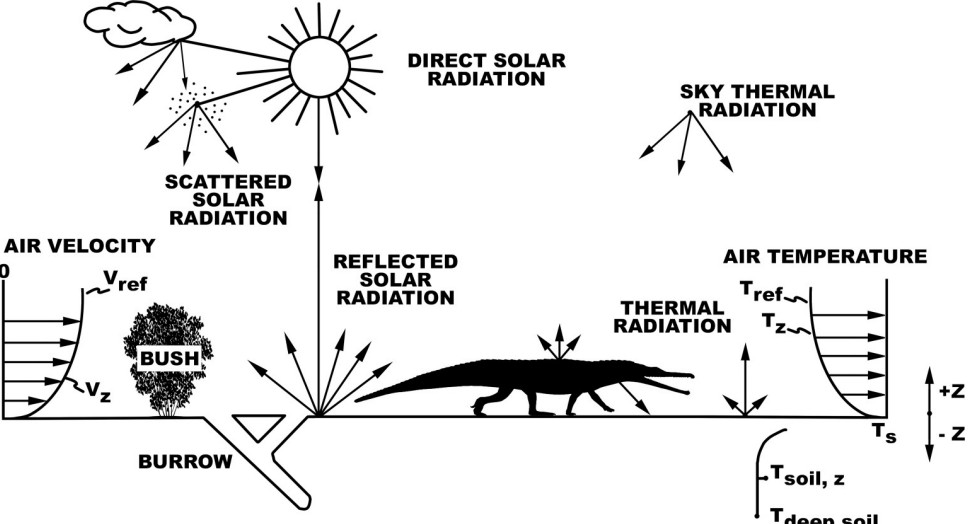

**Fig 1. Organism-environment heat balance interactions.** The reptile's heat and mass balances that influence body temperature are determined by where it chooses to be each hour to remain within its preferred body temperature range. Niche Mapper allows it to find a location each hour where it can remain active, or not, if necessary, to optimize its body temperature and/or water balance.

for paleotemperature and precipitation [52–54], global climate models [55], and global geo-chemical compilations [56,57]. We used these data to constrain our microclimate model to best represent a tropical Late Triassic environment and considered this the 'hot' microclimate model. This is not the most extreme temperatures proposed for the Late Triassic Chinle Formation [58], making our modeled environment a conservative estimate for testing whether high temperatures excluded large dinosaurs from the region. Our cold microclimate is an conservative approximation of paleoclimates in upper Triassic strata in central and northern Europe (~35–45˚N; [55,59,60]) where *Plateosaurus* is a relatively common constituent in fossil

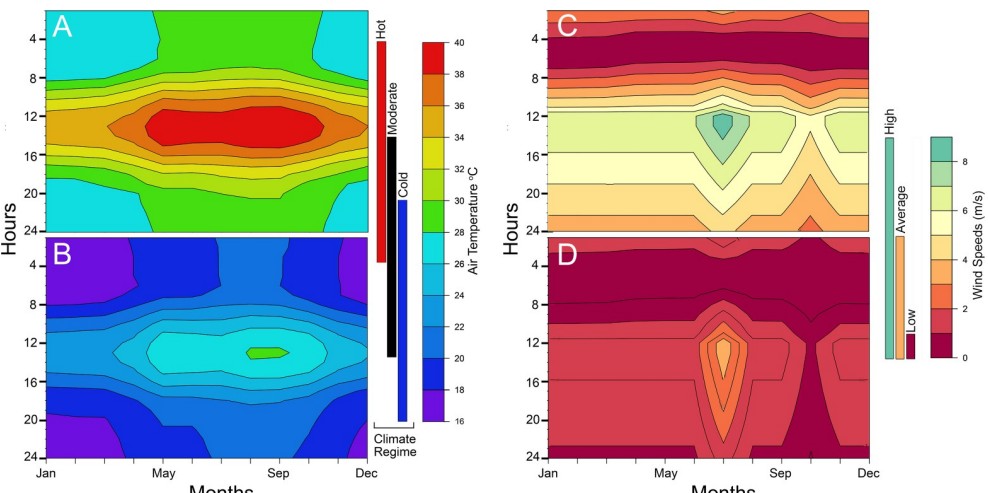

**Fig 2. Heatmaps of microclimate air temperature and wind speed at average animal height for each modeled hour.** We use a turbulent velocity and temperature profile where the most significant changes occur within the first 15 cm from the ground. The microclimates are the same for both *Coelophysis* (shown) and *Plateosaurus*. (A) hot and B) cold climate regimes, C) high and D) low wind speeds.

assemblages. In order to aid comparison and avoid interactions of variables, annual distribution of microclimate data was maintained across the hot and cold microclimates and only the temperature values were adjusted to model a cooler temperate microclimate as a first-order approximation of higher latitudes. A moderate microclimate was also modeled to represent areas intermediate between hot and cold microclimates. Air temperature are modeled at the average height of the organisms being examined for each hour (Fig 2). Paleolatitude is modeled as 12°N, with an elevation of 150 m [54].

Because the resolution of paleoclimate proxies are time-averaged it is difficult to determine an annual pattern for the microclimate model. As such, historic climate data from modern analogues provide a means to establish realistic annual patterns otherwise indiscernible in the rock record. We selected regions similar to paleogeographic reconstructions of the Chinle formation with respect to elevation, temperature and precipitation regimes, latitude, and general position on the continent; two locations in western Africa (Tamale, Ghana and Timbuktu, Mali) act as modern analogues for this study. Numeric values for the microclimate model were determined by multi-proxy data in the geological record whenever possible (see Table 1).

## The biophysical model

Gross organismal morphology (head, neck, torso, legs, tail) are modeled as simple geometric shapes (e.g., cylinders, spheres, truncated cones, or ellipsoids; S3 Appendix). These geometries have known or measurable heat transfer properties and temperature profile equations that simplify solving the heat balance equation when there is distributed internal heat generation [e.g., 36,64,65]. Each body part can be modeled with up to three concentric layers: 1) a solid central geometry of tissue uniformly generating heat; 2) if present, a surrounding layer of insulating subcutaneous fat is modeled as a hollow heat conducting geometry; 3) a surrounding layer of insulating fur or feathers, modeled as a hollow porous medium (see below). Net metabolic heat produced by the central flesh layer must be transferred through the fat layer to the skin surface, where is it either dissipated via cutaneous evaporation, convection and infrared thermal radiation (naked), or transferred through the fur/feather layer if present, then lost by convection and infrared thermal radiation to the environment. Heat is transferred through the fur/feathers by parallel thermal radiation and conduction through the air between the insulation elements and through the fur or feathers [36,65] and is lost to the environment via thermal radiation and convection. If the animal is lying down and has ventral insulation, it is compressed by an amount defined by the user and heat is conducted to or from the substrate at the insulation surface. Heat from solar radiation can also be absorbed through the skin (naked) or fur/feather layer (insulated), contributing to the heat load that must be dissipated (Fig 3).

Provided with the local environmental conditions from the microclimate model and biophysical properties of the organism, the animal model calculates radiative ($Q_{rad}$), convective ($Q_{conv}$), solar ($Q_{sol}$), and evaporative (respiratory: $Q_{resp}$ and cutaneous: $Q_{evap}$) heat fluxes between the animal and its microenvironment to solve a heat balance Eq (1) for a metabolic rate, $Q_{met}$, that satisfies (1) and is consistent with the status of its core and skin temperatures and environmental conditions:

$$Qmet - Qresp - Qst - Qevap = Qrad + Qconv - Qsol \qquad (1)$$

If the animal has a fur or feather layer, an additional parameter ($Q_{fur}$) must be added to account for heat flow through the insulating fur or feather layer:

$$Qmet - Qresp - Qs - Qevap = Qfur = Qrad + Qconv - Qsol \qquad (2)$$

**Table 1. Microclimate parameters inferred from geologic proxies, GCM's, and modern analogues.**

| Parameter | Model | Source | Modeled Range |
|---|---|---|---|
| Air Temperature | microclimate | [52] | Cold: 16–30˚C; warm: 20–34˚C; hot: 26–40˚C; **see Fig 2** |
| Relative Humidity | microclimate | [55,61] | 'Dry' 13–65%; 'Wet' 48–96% |
| Cloud Cover | microclimate | [61] | 50–90% |
| Wind Speeds | microclimate | [62,63] | 1–4 m/s; **see Fig 2** |
| Atmospheric %$O_2$ | biophysical | [56] | %$O_2$ = 18 |
| Atmospheric %$CO_2$ | biophysical | [54,57] | %$CO_2$ = 0.13 |

where $Q_{fur}$ represents the heat flux through the fur or feather layer via parallel conductive and radiative processes. For more detailed explanations of heat flux through porous media such as fur or feathers and solving for steady state conditions see Porter et al. [36], Porter and Kearney [66] and Mathewson and Porter [17].

For each hour of every model day the heat balance equation is solved for individual body parts and summed to provide the total metabolic rate (W) for the entire animal that will allow it to maintain a target core temperature in that hour's range of environmental conditions.

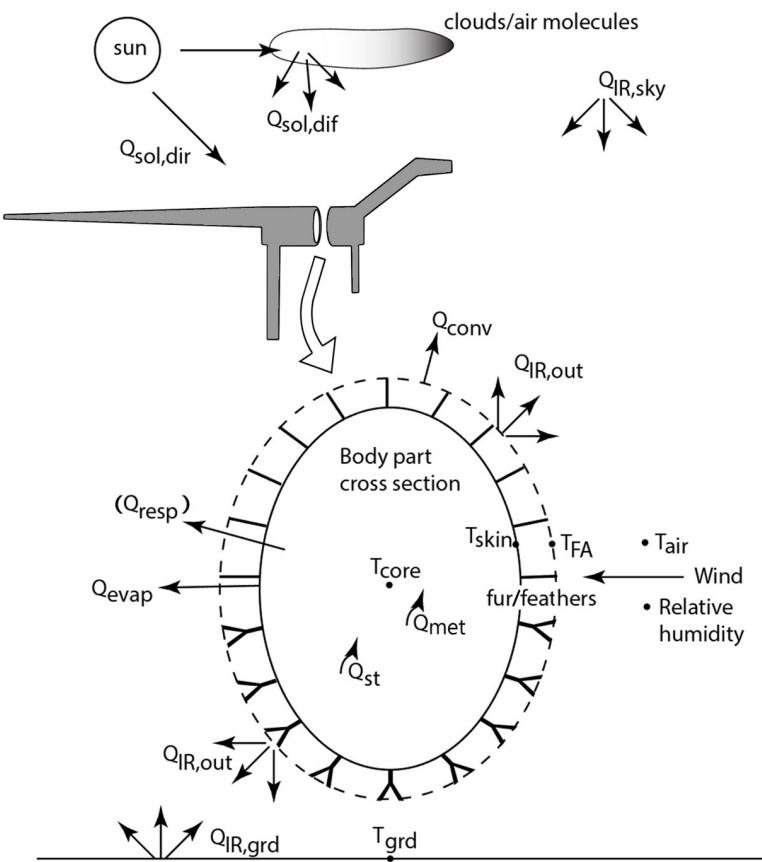

**Fig 3. Heat transfer pathways between modeled organism and environment.** Cross-section of a body segment (e.g., elliptical cylindrical torso of distributed heat generating flesh surrounded by an optional layer of fat (not shown), then skin surrounded by porous insulation whose properties may be the same or different dorsally vs. ventrally). The flesh is generating metabolic heat throughout the body (*Qmet*) and exchanging (*Qresp, Qevap, QIRnet, Qconv, Qsol*) heat with its environment as modeled by Niche Mapper (adapted from Porter et al. [23]). The transient model also includes a heat storage term, *Qst*, for the flesh. A full list of symbols and abbreviations can be found in the text.

Users can specify a basal metabolic rate multiplier to simulate activity in the heat balance calculations, as well as muscle efficiency, which is the proportion of that additional activity expenditure contributes to the animal's heat balance (i.e., 0% means that the mechanical work (activity) is 100% efficient with no excess heat produced; 99% means that 99% of the metabolic effort is lost as heat and needs to be considered in the heat balance). We assumed a mammal-like 20% muscle efficiency for activity with 80% of the chemical energy for activity going to heat. Although a ~35% muscle efficiency may be more reasonable for archosaurs, it is less well documented [67]; we performed a sensitivity analysis to test the effect of differing muscle efficiencies (see below).

If the total animal metabolic rate deviates from user-specified variation in the target metabolic rate (i.e., expected basal metabolic rate x activity multiplier) for the hour being modeled, physiological options, followed by behavioral options, are engaged to prevent the animal from being too hot or too cold by decreasing or increasing metabolic expenditure on heat production respectively.

User selected physiological options are engaged in the following order when individually enabled: 1) incrementally erect fur or feathers to increase insulation; 2) incrementally increase or decrease flesh thermal conductivity, simulating vasodilation or vasoconstriction of peripheral blood vessels; 3) incrementally increase or decrease core temperature, simulating temporary, bounded positive or negative heat storage; 4) incrementally increase the amount of surface area that is wet to increase evaporative heat loss, simulating sweating (if allowed); and 5) incrementally decrease oxygen extraction efficiency to increase respiratory heat loss, simulating panting.

If physiological changes are not sufficient to thermoregulate, behavioral thermoregulation options are engaged and the animal can seek shade, swim, wade, climb, or enter a burrow to achieve cooler environmental conditions. The user defines which behaviors are possible for the modeled organism; for instance, it is unlikely that an 850 kilogram prosauropod is burrowing or climbing trees to behaviorally thermoregulate, so these options would not be utilized. If the animal is too cold (i.e. the requisite metabolism is greater than the resting metabolic rate x the activity multiplier), the user can allow the animal to enter a burrow or seek vegetative shelter or get out of the wind. These options also reduce radiant heat loss to the sky by providing overhead structures (e.g. forest canopy or burrow ceiling) with warmer radiant temperatures than the open sky. Users can also allow model animals to make postural changes such as curling up to minimize surface area for heat exchange with the environment if the animal is too cold.

The heat balance is re-solved after each incremental thermoregulatory change until either 1) the metabolic rate that balances the heat budget equation is within the percent error of the target metabolic rate or 2) thermoregulatory options are exhausted. In the case of the latter, the metabolic rate that balances the heat budget equation that is closest to the target rate is used for that hour. Hourly metabolic rates and water losses are integrated over the day to calculate daily metabolic rate and water loss, which can then be used to calculate food and water requirements. The day's water and energy requirements are then used to compute the respiratory and digestive system inputs and outputs using molar balances as described below.

The heat and mass balance of an organism are connected by metabolic rate, a 'biological fire' that requires fuel and oxygen. The daily metabolic rate that releases heat ($Qmet$) sets the daily mass balance requirements for the respiratory and digestive systems (Fig 4). Diet composition (proteins, carbohydrates, lipids, percent water) specify how much mass must be absorbed ($\mathbf{m_{abs}}$) from the gut to meet metabolic demands. Digestive efficiency divided into the mass absorbed determines the mass of food that must be ingested per day ($m_{in}$) to meet energy requirements. Mass excreted ($m_{out}$) is the difference between mass in and mass absorbed.

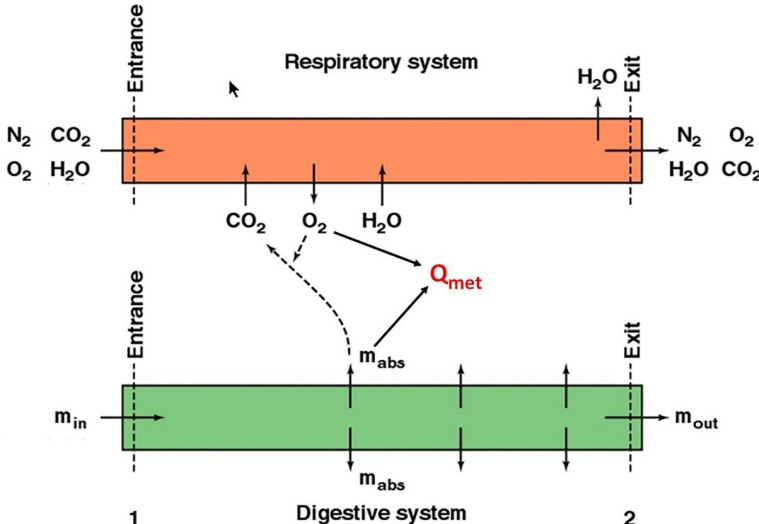

**Fig 4. Internal mass balance models coupled to heat transfer.** System diagram for the respiratory and digestive system driven by the metabolic rate, *Qmet*.

The respiratory system functions in an analogous manner. Diet utilization and activity rates determine the amount of oxygen needed and carbon dioxide produced. Oxygen required divided by the respiratory extraction coefficient specifies the mass of air that must enter the respiratory system, given the atmospheric composition of oxygen per unit volume. Humidity of the incoming air is increased to saturated air at lung (body) temperature, so respiratory water loss can also be computed. Recovery of water vapor during exhalation through cooler nostrils is also calculated.

**Biophysical model parameterization in deep time.** Key *morphological* and *behavioral* model inputs are summarized in Tables 2 and 3. For *behavioral* thermoregulation, we allowed animals to hide from wind (if too cool), seek shade during the day (if too hot), seek shade at night (e.g., simulating vegetative cover slowing the rate of radiative heat loss seen in open sky conditions), be active in the shade (day and night), postural changes to minimize surface area if cold and inactive, or maximize surface area if hot. For *physiological* thermoregulation we

**Table 2. Morphological parameters of porous insulation modeled and skin or insulation surface solar reflectivity.**

| Variable | | | No Insulation | Top-only Insulation | Full Insulation | Reference and comment |
|---|---|---|---|---|---|---|
| Insulation | Length (mm) | Dorsal* | 0 | 30 | 30 | [Based on 69–72] |
| | | Ventral* | 0 | 0 | 30 | |
| | Depth (mm) | Dorsal* | 0 | 10 | 10 | |
| | | Ventral* | 0 | 0 | 10 | |
| | Density (elements/cm$^2$) | | 0 | 2000 | 2000 | [37 Fig 25,73] |
| | Insulation/skin reflectivity (%) | | 15 | 15 | 15 | Porter, marine iguana, unpub. data |
| | Pilo/ptiloerection | | No | Yes** | Yes** | |

Insulation element density estimated assuming that 1) feathers evolved by selection for a follicle that would grow an emergent tubular appendage [73] early 'feathers' were similar to stage 1 or 2 in the developmental pattern of modern feathers, i.e. cylindrical in shape. 3) that the minimum hair density that substantially reduces heat loss (reduces metabolic heat generation that sets body temperature) is approximately 800–1000 'hairs' /cm$^2$ [37 Fig 25].

* head, neck, torso, and tail

**only when behaviors are enabled

**Table 3. Parameters for metabolic rates, diet, and behavior.**

| Variable | | | Value | | Reference and comment |
|---|---|---|---|---|---|
| | | | *Plateo* | *Coelo* | |
| Squamate | Metabolic rate | | 70.1 W | 3.4 W | [7] |
| | Core temp (˚C) | Maximum | 40 | | |
| | | Target | 38 | | |
| | | Minimum | 25 | | |
| Ratite | Metabolic rate | | 301 W | 24 W | [74,75] |
| | Core temp (˚C) | Maximum | 40 | | [76] |
| | | Target | 38 | | [76] |
| | | Minimum | 36 | | |
| Fat mass as % of body mass | | | 5 | | [74,75] |
| Sweat ok? | | | No | | |
| Activity Multiplier** | | | 2 | | [77] |
| Food | High Browser | % protein* | 10 | | [78] |
| | | % fat* | 0 | | |
| | | % carbohydrate* | 88 | | |
| | | % dry matter | 25 | | |
| | | Digestive efficiency (%) | 60 | | |
| | | Fecal water (%) | 60 | | [79] |
| | | Uric acid (%) | 60 | | [79] |
| | Low Browser | % protein* | 10 | | [78] |
| | | % fat* | 0 | | |
| | | % carbohydrate* | 88 | | |
| | | % dry matter | 44 | | |
| | | Digestive efficiency (%) | 60 | | |
| | | Fecal water (%) | 60 | | [79] |
| | | Uric acid (%) | 60 | | [79] |
| | Carnivore | % protein* | 60 | | [80] |
| | | % fat* | 38 | | |
| | | % carbohydrate* | 0 | | |
| | | % dry matter | 66.7 | | |
| | | Digestive efficiency (%) | 85 | | |
| | | Fecal water (%) | 46 | | [81] |
| | | Uric acid (%) | 27 | | [81] |
| Behavior | Ground seeking shade | | Y | Y | |
| | Night shade | | Y | Y | |
| | Seek shelter from wind | | Y | Y | |
| | Posture | Sleeping | Posture 3 | | Legs lumped into torso, head & neck on ground |
| | | Resting in shade | Posture 3 | | Legs lumped into torso, head & neck on ground |
| | | Inactive posture | Posture 3 | | Legs lumped into torso, head & neck on ground |

'*Plateo*' = *Plateosaurus* and '*Coelo*' = *Coelophysis*.

* based on % dry mass

** Activity multiplier = field metabolic rate (FMR) / basal metabolic rate (BMR). FMR is calculated as FMR = 4.82*weight^(0.734)

allowed for panting (too hot), increased and decreased flesh conductivity (to simulate vasocon-striction/vasodilation when cold/hot), if dermal insulation (fur/feathers) is present they can piloerect/ptiloerect, or changes in regulated body temperature within the user specified

maximum and minimum body temperature range. Sweating was not enabled due to phyloge-
netic constraints. Sensitivity analyses were performed to test which behavior, or interaction of
multiple behaviors, had the strongest effect. Since legs and tail consist of mostly muscle, bone
and tendon, we allowed the temperature in limb and tail segments to reach 50% of the differ-
ence between the torso-segment junction and ambient air or ground temperatures [c.f.,12,68].

**Determining rates of metabolism.** To determine a range of metabolic rates for extinct
taxa within our modeled microclimate, we simulated a spectrum of different metabolic rates.
We evaluated 5 different resting metabolic rates (RMRs) ranging from a typical ectothermic
squamate to endothermic eutherian metabolisms. An RMR from the lower avian range (ratite)
and two lower mammal (monotreme and eutherian) RMRs were calculated from empirical
regressions utilizing phylogenetic and ecological constraints [e.g.,74,75], while the squamate,
and an additional eutherian RMR were calculated using empirical models derived from oxygen
consumption or $CO_2$ production measurements with an assumed respiratory quotient [7,82].
These regressions implicitly include the presence or absence of epidermal insulation of extant
species as well as their size and shape, but the data provide a range of values for estimating the
span of modern mass-specific metabolic rates(S1 Table).

Using the calculated masses (S3 Appendix) and the empirical equations above we generated
five different mass-specific resting metabolic rates for *Coelophysis* and *Plateosaurus*, labeled:
squamate, monotreme, marsupial, ratite, and eutherian. We analysed the thermoneutral range
for each RMR in a virtual metabolic chamber simulation within Niche Mapper. We chose to
conduct the remainder of the model simulations with low (squamate), moderate (monotreme),
and high (ratite) RMRs representing a possible range of metabolic rates based on phylogenetic
position. It is unlikely that basal dinosaurs had metabolic rates elevated above extant ratites, or
below extant squamates.

**Diets.** In the Niche Mapper model, the primary outputs influenced by diet are daily values
for discretionary water (kg/day) and food requirement (kg/day). The diet (required caloric
intake) is calculated based on daily energy expenditure, user-supplied values for percent pro-
tein, fat, carbohydrates, and dry mass of the food, and the animal's assimilation efficiency
(Table 3). The amount of water initially available to the organism is calculated from the diet-
assigned dry mass and the amount of food consumed (e.g., total food mass—dry mass = total
available free water from food). Metabolic water production is computed from diet composi-
tion and metabolic rate [83 p. 489, 695]. Daily water loss is the total of cutaneous water evapo-
ration (if sweating were allowed; we did not allow sweating) and water lost through respiration
and excretion (in mammal-based models this includes water loss through feces and urine, in
non-mammal models water loss through urine is ignored). If the daily water budget is nega-
tive, the organism must drink water to make up the volume; thus, discretionary water reflects
the debt or credit of the total water budget after all modeled physiological needs are met. A
user defined digestive efficiency controls the amount of incoming calories (food mass) that is
required by the model organism to meet its metabolic needs (Table 3).

*Coelophysis* has long been considered a predatory theropod [84,85] and *Plateosaurus* is usu-
ally described as a herbivorous prosauropod, but omnivory is not excluded [6]. To assess the
impact of varying inferences of diet on food and water requirements *Coelophysis* and *Plateo-
saurus* were both modeled as carnivores and as high and low browsing herbivores. The diet in
the high browsing scenario is comprised of primarily high % dry matter (e.g., 40–50% dry
mass) such as conifers (8.3 MJ/kg dry mass), ginkgos (8.6 MJ/kg dry mass), and cycads (6.1
MJ/kg dry mass) [78,86]. Low browsing diets were primarily composed of ferns (7.7 MJ/kg dry
mass) and *Equisetum* (11.6 MJ/kg dry mass) which contain much higher water content (e.g.,
25–30% dry mass) [78,86]. A positive discretionary water budget would indicate the animal is

getting most of its water from its food source as well as metabolic water and may not require regular access to drinking water extending its potential geographic range.

**Energy requirements.** We developed an R script to interface Niche Mapper with a modifiable database containing climate (Table 1) and physiological variables (Tables 2 and 3) for each of the experimental simulations, which were assessed for 6 unique climates [hot, moderate, cold] x [arid, humid]. For each simulation Niche Mapper calculated hourly interactions between the organisms and their environment over a 24 hour period at mid-month for a calendar year (12 total model days).

In all of our modeled simulations, each species is given the potential ability to be active every hour of the day (24 hours). The amount of metabolic heat production (W) needed to maintain the target core temperature throughout the day is determined by multiplying the resting metabolic rate (Table 4) by an activity multiplier (2.0 in our study; [77]). The resultant is the daily target metabolic rate (MJ/day):

$$RMR(W) \; x \; 2.0 \; x \; 86,400(s/day)/1.0x10^6 \; J/MJ \tag{3}$$

Thus any decrease in activity hours represents periods of the modeled day when the animal is heat stressed and must decrease activity to lower its metabolic heat production. If the animal is cold stressed or within its active thermoneutral zone it will maintain 24 hour activity. However, if it is cold stressed the animals metabolic heat production and by extension, food consumption, must increase accordingly.

We define the *active* thermoneutral zone as the zone where an activity multiplier > 1 is expanding the temperature range in which the animals internal heat production balances the heat loss to the external environment (steady state condition). In contrast, a *resting* thermoneutral zone is the temperature range when the activity multiplier is 1.

Four physiological conditions were used to test the viability of each modeled organism under six microclimate conditions mentioned above (see Table 1). Low (squamate) and high (ratite) resting metabolic rates were calculated based on equations from McNab [74,75] and McMahon [7], each of which were analyzed with a broad squamate-like core temperature range (CTR), which ranged from 26–40˚C, moderate monotreme-like CTR (32–40˚C), and a narrow ratite-like CTR (36–40˚C). All CTRs were assigned a target core temperature of 38˚C.

## Metabolic chamber

Metabolic chamber simulations in Niche Mapper were used to evaluate the specific impact of different physiological inferences and their impact on the temperatures in which model animals were predicted to be cold or heat stressed. In the metabolic chamber simulations, all temperatures (ground, sky, and air) are set equal to one another, no solar input is allowed, a constant, negligible wind speed of 0.1 m/s is used, along with a constant 5% relative humidity. Animals are modeled "at rest" in a standing posture with no activity multiplier. In order to identify lower and upper critical temperature boundaries for each animal, heat balance calculations were performed along a range of temperatures (0–51˚C) that exceeded the minimum and maximum air temperatures within which organisms could maintain thermoneutrality [87]. This process was repeated for each proposed metabolic rate.

## Sensitivity analyses

Niche Mapper is an effective tool for modeling extant organisms where direct measurements can be applied. Modeling organisms in deep-time is faced with a number of challenges where direct measurements are not possible. Variables such as air temperature, core temperature

**Table 4. Annual predicted energy budget (MJ/year) for both dinosaur species.**

| | Mass (kg) | Resting Metabolic Rate (W) | | | Daily Metabolic Rate (MJ) | | | Annual Metabolic Rate (MJ) | | |
|---|---|---|---|---|---|---|---|---|---|---|
| | | Ecto RMR* | Mono RMR** | Ratite RMR** | Ecto DMR | Mono DMR | RatiteDMR | Ecto YMR | Mono YMR | Ratite YMR |
| Coelo | 15 | 2.6 | 9.7 | 17.5 | 0.45 | 1.67 | 3.02 | 163.08 | 610.29 | 1102.58 |
| | 21 | 3.4 | 12.3 | 22.2 | 0.59 | 2.13 | 3.83 | 214.36 | 778.64 | 1397.75 |
| | 30 | 4.5 | 16.0 | 28.5 | 0.79 | 2.76 | 4.92 | 286.62 | 1008.05 | 1797.36 |
| | 40 | 5.7 | 19.7 | 34.9 | 0.99 | 3.40 | 6.03 | 362.43 | 1241.48 | 2201.49 |
| | 50 | 6.9 | 23.1 | 40.9 | 1.19 | 4.00 | 7.06 | 434.87 | 1459.16 | 2576.55 |
| | 60 | 8.0 | 26.4 | 46.5 | 1.38 | 4.56 | 8.03 | 504.72 | 1665.06 | 2929.96 |
| Plateo | 600 | 52.7 | 139.8 | 235.5 | 9.11 | 24.16 | 40.70 | 3325.34 | 8819.21 | 14854.63 |
| | 850 | 70.1 | 179.9 | 301.1 | 12.12 | 31.09 | 52.03 | 4422.46 | 11348.73 | 18989.15 |
| | 1150 | 89.9 | 224.0 | 372.6 | 15.53 | 38.70 | 64.38 | 5668.06 | 14125.16 | 23499.42 |
| | 1600 | 117.8 | 284.4 | 470.3 | 20.36 | 49.15 | 81.26 | 7430.36 | 17940.35 | 29659.90 |
| | 2000 | 141.5 | 334.3 | 550.4 | 24.44 | 57.77 | 95.10 | 8921.95 | 21085.98 | 34712.93 |
| | 3000 | 197.2 | 448.4 | 732.5 | 34.08 | 77.48 | 126.57 | 12440.32 | 28280.29 | 46199.32 |

See text for explanation of under and over estimates of mass (kg). Green highlighted mass and data are our chosen optimal masses for each species.

* [7]

*[74,75]

range, resting metabolic rate, and insulatory structures have a significant effect on the modeled organisms annual metabolic energy and are tested for and visualized in each modeled experiment. Additional parameters, such as muscle efficiency, digestion, and respiration, as well as mass estimates, skin reflectivity, and insolation factors related to latitude are independently tested for model sensitivity. In order to determine how sensitive the model was to these additional parameters, a bounded range that includes our hypothesized values were modeled for each parameter.

For instance, there is uncertainty as to the color of skin or insulatory structures in most extinct animals. For our purposes, it is known that a lighter color absorbs less solar radiation, a darker color absorbs more and this variable could be easily selected for in a given environment [88–90]. To test the effect of color on total metabolic energy requirements we modeled the skin of *Plateosaurus* and the uninsulated *Coelophysis* as well as the proto-feathers for the top-only and fully insulated *Coelophysis* with 5 states ranging from high to low reflectivity (light to dark color, respectively) in both our cold and hot microclimate.

Similarly we tested main and interactive effects of parameters related to climate (i.e.,temperature, wind, relative humidity, cloud cover). To assess the relative effect of these parameters we used the metric of total annual energy for each species and determined how annual metabolic expenditure would change relative to the target value for all variables individually and combined. This approach allowed us to evaluate the main effects of each of these variables as well as possible interactions between them. To determine main effects and interactions of the variables on annual metabolic expenditures for *Plateosaurus* and *Coelophysis* we used a $2^4$ (climate) full factorial design and Yates' algorithm for analysis of effects [91]; minimum and maximum data are outlined in Table 1.

## Results

### Sensitivity analyses

The strength of our modeled results, in part, relies on understanding how the model responds to ranges of values for variables that are not directly measurable in deep-time. We conducted

the following analyses to quantify the advantage or disadvantage our chosen values would impact on the model results: skin/insulation reflectivity, muscle efficiency, respiratory efficiency, digestive efficiency, the effect of latitude, and mass estimates; a summary of these results follows—further details and figures are provided in supplemental data (S4 Appendix).

Skin and insulation color (reflectivity) was analyzed from 10–60% (15% was our chosen model value). It was observed that the disparity in ME between high and low reflectivity values increased with increasing cold stress. For example, the more cold-stressed the model was (i.e., >4–5 x RMR), the greater advantage low reflectivity values (darker color) had. However, there was a negligible effect of reflectivity for models whose annual ME was near target (e.g., between 2x and 3x RMR). Similarly, muscle efficiency ranged from 20–50% efficient (20% was our chosen model value) and the disparity between the lowest and highest values for a given model increased as cold-stress increased (see S4 Appendix); there was a negligible effect for models whose RMR was between 2 and 3x resting. An analysis of respiratory efficiency ranged from 10–30% (we chose a min-max value of 15–20%) and there was virtually no change in annual ME regardless of which parameter was used.

Digestive efficiency was analyzed with a range of 70–85% efficiency (we chose 85%) for the carnivorous diet, and 30–70% (we chose 60%) for the herbivorous diets (see S4 Appendix for discussion). All diet parameters are independent of metabolic calculations and thus did not affect annual ME. Varying the efficiency of digestion provides us with a range of annual wet food requirements. These values can be used to compare with reasonable rates of browsing (or prey acquisition/consumption) relative to modern analogs. For instance, even at the lower extreme, a 30% digestive efficiency for *Plateosaurus* would require ~8000 kg wet-food per year; this is ~22 kg of wet-food per day, which is on par with similarly sized extant browsing mammals such as the black rhinoceros [92,93].

Because we are using our cold microclimate as a proxy for higher latitudes we also tested our models at 45˚N [e.g., 55,59,60]. The primary effect of increasing latitude was a result of increased daylight hours midyear and decreased daylight hours during the winter months. This is most apparent in the increased hours/day that core temperature was maintained, mid-year, and decreased during the winter months relative to those observed at 12˚N. The model is more sensitive to microclimate temperatures than variance in insolation due to increased latitude between12 and 45˚N.

The parameters outlined above had relatively small effects on metabolic needs of the modeled organisms that were able to maintain an annual ME between 2 and 3x RMR. However, we realize these effects can be cumulative and are more significant at the boundaries of a modeled organisms' temperature tolerance where small changes can be the difference between survival or death. There is also the potential effect of interaction between parameters. To test the main and interactive effects of four primary climate parameters (temperature, humidity, wind speed, and cloud cover) a $2^4$ factorial design and Yates' algorithm [91] was analyzed (S4 Appendix Fig 6). Temperature had 2–10 times the effect of wind, and both humidity and cloud cover were insignificant. Variables that have the greatest impact, such as temperature, CTR, RMR, and insulation are presented below with a range of inputs for each experiment.

## Mass estimates

Mass estimates can vary widely for a given taxon depending on methodology [94–99]. Niche Mapper uses a user-supplied mass and distributes that mass, with assigned densities, among each body segment (head, neck, torso, front legs, hind legs, and tail). We tested our modeled organisms with 6 mass estimates from a low to an extreme high mass in both the hot and cold

microclimate. In addition to increasing the estimated mass we also accounted for the necessary increase in RMR with mass (see Table 4).

Linear measurements of a *Coelophysis* specimen (AMNH 7224) yield a skeletal length of 2.61 m. With an estimated average density of 0.97 kg/l our *Coelophysis* model has a mass of 21 kg. This is consistent with previously assigned masses [94, pg. 260] of 15–20 kg for a 'gracile' and 'robust' skeleton, respectively. It is unlikely that the mass exceeds 30 kg for the skeletal dimensions used for this analysis of mass estimates. We tested the effect of increased mass via increasing the diameter of the model's body segments (i.e., making it thicker) with mass assignments of 15, 21, 30, 40, 50, and 60 kg (see S4 Appendix Fig 7). The 40, 50, and 60 kg masses are extreme (nonviable) overestimates to test the model. Results for monthly metabolic energy with a high (ratite-like) RMR and CTR (relative to target) for the uninsulated, top-only insulated, and fully insulated *Coelophysis* model demonstrate the effect of mass and insulation in hot and cold microclimates (Fig 5). Results for varied RMR and CTR for the 21 kg *Coelophysis* follow.

The uninsulated model resulted in extreme cold-stress for the lower three sizes in the hot microclimate, and all 6 mass estimates in the cold microclimate. When increased to top-only insulation the severity of cold-stress decreased, but the model was still excessively cold-stressed in the cold microclimate. The hot microclimate resulted in more months where the model was able to maintain its target metabolic energy, including summer months of the lower three size estimates and all months for the largest three mass estimates. The fully insulated model shows heat-stress during the summer months in the hot microclimate, gradually increasing in severity and temporal extent with increased mass. Under the cold microclimate, the model met target metabolic energy for all but the 15 kg *Coelophysis*, which exhibited minor cold-stress within the winter months (between 5 and 10% above target).

Our linear dimensions for *Plateosaurus* were taken from GPIT/RE/7288, a six meter long skeleton with a femoral length of 635 mm. With an estimated density of 0.97 kg/l the model has a mass of 850 kg. This is in line with mass estimates for moderate sized *Plateosaurus* specimens (i.e., 5.67 m skeleton [595 mm femoral length] with a mass between 660–782 kg using a 0.89–1.05 kg/l density respectively; [98]). Other *Plateosaurus* mass estimates include a 6.5 m long skeleton (1073 kg using polynomial method [99]) and a 920 kg mass determined by stylopodial circumference using a 685 mm femoral length [100].

To test the effect of different mass estimates, we chose to take the same skeletal dimensions and increase or decrease the diameter of body segments (assigned densities remained constant). We ran experiments assuming a total mass of 600, 850, 1150, 1600, 2000, and 3000 kg (Fig 6; see also S4 Appendix Fig 8). The first three states (600–1150 kg) more likely capture a realistic mass estimate range for the skeleton and are representative of mass estimates in the literature for this specimen [97–100]. The last three states (1600–3000 kg) were used to observe how an extreme (nonviable) overestimate of mass would affect the model results. Results for monthly metabolic energy with a high (ratite-like) RMR and CTR (relative to target) for the *Plateosaurus* model demonstrate the effect mass has for hot and cold microclimates (Fig 6). Results for varied RMR and CTR for the 850 kg *Plateosaurus* follow.

The 600 kg model was mildly heat stressed in the hot microclimate during peak summer temperatures, however it was excessively cold (ME ~15–20% above target) in the cold microclimate. Under the hot microclimate, the 850 kg model we identified as most likely for our 6 m skeleton met its target ME during the cooler winter and spring/fall seasons, but experienced significant heat stress (ME ~10% below target) during peak summer temperatures. As mass increased, this trend was amplified in the hot microclimate producing excessive heat stressed models. The 850 kg model experienced modest cold stress in the winter months, while the four largest mass estimates met expected target values within the cold microclimate.

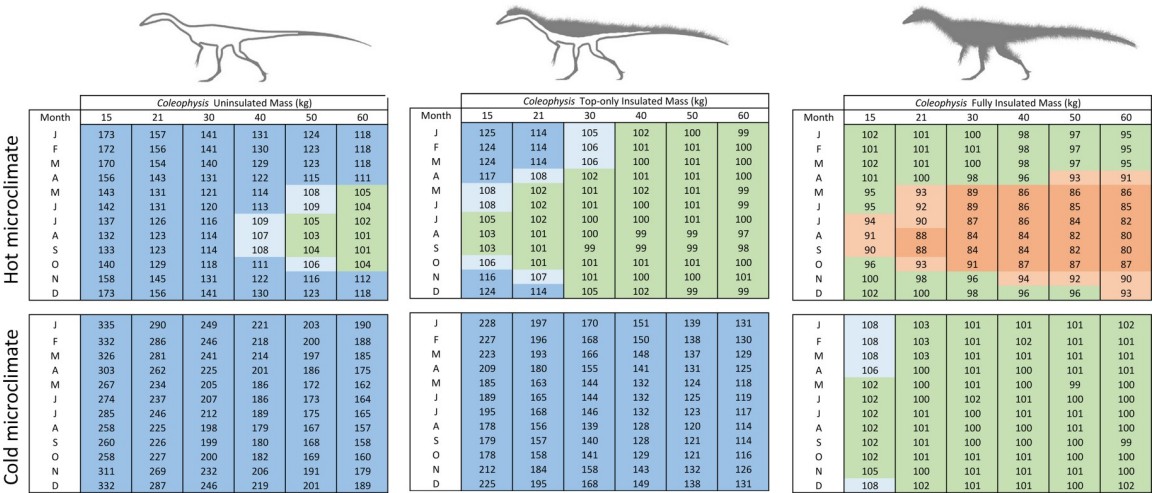

**Coleophysis Uninsulated Mass (kg)**

| Month | 15 | 21 | 30 | 40 | 50 | 60 |
|---|---|---|---|---|---|---|
| **Hot microclimate** | | | | | | |
| J | 173 | 157 | 141 | 131 | 124 | 118 |
| F | 172 | 156 | 141 | 130 | 123 | 118 |
| M | 170 | 154 | 140 | 129 | 123 | 118 |
| A | 156 | 143 | 131 | 122 | 115 | 111 |
| M | 143 | 131 | 121 | 114 | 108 | 105 |
| J | 142 | 131 | 120 | 113 | 109 | 104 |
| J | 137 | 126 | 116 | 109 | 105 | 102 |
| A | 132 | 123 | 114 | 107 | 103 | 101 |
| S | 133 | 123 | 114 | 108 | 104 | 101 |
| O | 140 | 129 | 118 | 111 | 106 | 104 |
| N | 158 | 145 | 131 | 122 | 116 | 112 |
| D | 173 | 156 | 141 | 130 | 123 | 118 |
| **Cold microclimate** | | | | | | |
| J | 335 | 290 | 249 | 221 | 203 | 190 |
| F | 332 | 286 | 246 | 218 | 200 | 188 |
| M | 326 | 281 | 241 | 214 | 197 | 185 |
| A | 303 | 262 | 225 | 201 | 186 | 175 |
| M | 267 | 234 | 205 | 186 | 172 | 162 |
| J | 274 | 237 | 207 | 186 | 173 | 164 |
| J | 285 | 246 | 212 | 189 | 175 | 165 |
| A | 258 | 225 | 198 | 179 | 167 | 157 |
| S | 260 | 226 | 199 | 180 | 168 | 158 |
| O | 258 | 227 | 200 | 182 | 169 | 160 |
| N | 311 | 269 | 232 | 206 | 191 | 179 |
| D | 332 | 287 | 246 | 219 | 201 | 189 |

**Coleophysis Top-only Insulated Mass (kg)**

| Month | 15 | 21 | 30 | 40 | 50 | 60 |
|---|---|---|---|---|---|---|
| **Hot microclimate** | | | | | | |
| J | 125 | 114 | 105 | 102 | 100 | 99 |
| F | 124 | 114 | 106 | 101 | 101 | 100 |
| M | 124 | 114 | 106 | 100 | 101 | 100 |
| A | 117 | 108 | 102 | 101 | 101 | 100 |
| M | 108 | 102 | 101 | 102 | 101 | 99 |
| J | 108 | 102 | 101 | 100 | 101 | 99 |
| J | 105 | 102 | 100 | 100 | 101 | 100 |
| A | 103 | 101 | 100 | 99 | 99 | 97 |
| S | 103 | 101 | 99 | 99 | 99 | 98 |
| O | 106 | 101 | 101 | 101 | 101 | 100 |
| N | 116 | 107 | 101 | 100 | 100 | 101 |
| D | 124 | 114 | 105 | 102 | 99 | 99 |
| **Cold microclimate** | | | | | | |
| J | 228 | 197 | 170 | 151 | 139 | 131 |
| F | 227 | 196 | 168 | 150 | 138 | 130 |
| M | 223 | 193 | 166 | 148 | 137 | 129 |
| A | 209 | 180 | 155 | 141 | 131 | 125 |
| M | 185 | 163 | 144 | 132 | 124 | 118 |
| J | 189 | 165 | 144 | 132 | 125 | 119 |
| J | 195 | 168 | 146 | 132 | 123 | 117 |
| A | 178 | 156 | 139 | 128 | 120 | 114 |
| S | 179 | 157 | 140 | 128 | 121 | 114 |
| O | 178 | 158 | 141 | 129 | 121 | 116 |
| N | 212 | 184 | 158 | 143 | 132 | 126 |
| D | 225 | 195 | 168 | 149 | 138 | 131 |

**Coleophysis Fully Insulated Mass (kg)**

| Month | 15 | 21 | 30 | 40 | 50 | 60 |
|---|---|---|---|---|---|---|
| **Hot microclimate** | | | | | | |
| J | 102 | 101 | 100 | 98 | 97 | 95 |
| F | 101 | 101 | 101 | 98 | 97 | 95 |
| M | 102 | 101 | 100 | 98 | 97 | 95 |
| A | 101 | 100 | 98 | 96 | 93 | 91 |
| M | 95 | 93 | 89 | 86 | 86 | 86 |
| J | 95 | 92 | 89 | 86 | 85 | 85 |
| J | 94 | 90 | 87 | 86 | 84 | 82 |
| A | 91 | 88 | 84 | 84 | 82 | 80 |
| S | 90 | 88 | 84 | 84 | 82 | 80 |
| O | 96 | 93 | 91 | 87 | 87 | 87 |
| N | 100 | 98 | 96 | 94 | 92 | 90 |
| D | 102 | 100 | 98 | 96 | 96 | 93 |
| **Cold microclimate** | | | | | | |
| J | 108 | 103 | 101 | 101 | 101 | 102 |
| F | 108 | 103 | 101 | 102 | 101 | 101 |
| M | 108 | 103 | 101 | 101 | 101 | 101 |
| A | 106 | 100 | 101 | 100 | 101 | 101 |
| M | 102 | 100 | 101 | 100 | 99 | 100 |
| J | 102 | 100 | 102 | 101 | 100 | 100 |
| J | 102 | 101 | 100 | 101 | 101 | 100 |
| A | 102 | 101 | 100 | 100 | 101 | 100 |
| S | 102 | 101 | 100 | 100 | 100 | 99 |
| O | 102 | 101 | 101 | 101 | 100 | 100 |
| N | 105 | 100 | 101 | 101 | 101 | 100 |
| D | 108 | 102 | 101 | 101 | 100 | 102 |

**Fig 5. Effect of mass estimate (*Coleophysis*) on annual energy.** The matrix reflects the effect of size and insulation for *Coleophysis* in a hot and cold microclimate. Dark blue = >10% above target ME; light blue = 5–10% above target ME; green = +/- 5% of target ME; light orange = 5–10% below target ME; and dark orange = >10% below target ME.

## Diet of *Plateosaurus*

It has been suggested [87] that *Equisetum* would have been a favored food source (from a nutritional point of view) due to its higher degradability (e.g., 11.6 MJ/kg dry matter) relative to various conifers or *Ginko* (8.3, 8.6 MJ/kg dry mass, respectively). However, given the high water content of *Equisetum* (~70% [79]) relative to conifers (~44% [79]) the degradable energy per kilogram of wet mass (what the animal actually consumes) is nearly identical: 3.5 MJ/kg wet mass (*Equisetum*) vs 3.6 MJ/kg wet mass (various conifers) [79,87]. The various ferns reported by Hummel and others [87] have nearly 75% water content and yield 7.7 MJ/kg dry mass (or 2.1 MJ/kg wet mass). Thus, an animal eating dominantly ferns will need to consume 60% more vegetative mass than an organism whose diet is primarily composed of conifers or horsetails.

The diet component of the model, although extremely useful for certain questions, is calculated based on the resulting metabolic energy outputs. Factors such as digestive efficiency, food nutrient composition, waste products (urea/feces), and gut retention time affect the food and water requirements, but do not directly affect metabolic energy calculations. In the two modeled herbivorous diet scenarios the high-browsing animals display substantial differences in the volume of food required per day relative to low browsing animals. This is due to differences in the %dry mass, where the higher the %dry mass, the greater non-water component is available for digestion (see above, and Fig 8). *Plateosaurus*, as a high browser with ratite RMR and CTR in the cold microclimate meets the calculated target food intake (blue filled pentagon, Fig 8). The lower than target values for high browsing in the hot microclimate demonstrate a decrease in activity below 2 times RMR, likely due to heat stress.

## Diet of *Coelophysis*

The incremental addition of insulation to *Coelophysis* produced a corresponding decrease in overall food requirement. The uninsulated *Coelophysis* (ratite RMR/CTR) with a carnivorous diet in the hot climate requires ~300 kg/y, which is near the calculated target food intake requirement of 310 kg/y (blue filled pentagons of Fig 7). However, with full insulation the annual intake is only 200 kg/y, suggesting heat stress has an impact on activity through a

| Month | *Plateosaurus* Mass (kg) | | | | | |
|---|---|---|---|---|---|---|
|  | 600 | 850 | 1150 | 1600 | 2000 | 3000 |
| J | 99 | 99 | 98 | 94 | 94 | 84 |
| F | 99 | 99 | 98 | 94 | 94 | 84 |
| M | 100 | 99 | 98 | 94 | 94 | 84 |
| A | 99 | 98 | 97 | 88 | 89 | 78 |
| M | 96 | 92 | 89 | 77 | 79 | 68 |
| J | 95 | 92 | 89 | 77 | 79 | 68 |
| J | 95 | 92 | 88 | 75 | 77 | 65 |
| A | 92 | 89 | 85 | 71 | 73 | 62 |
| S | 93 | 89 | 85 | 71 | 73 | 62 |
| O | 95 | 93 | 89 | 76 | 78 | 68 |
| N | 99 | 97 | 95 | 86 | 88 | 77 |
| D | 100 | 98 | 97 | 92 | 93 | 82 |

| | | | | | | |
|---|---|---|---|---|---|---|
| J | 119 | 107 | 100 | 100 | 100 | 98 |
| F | 118 | 106 | 99 | 100 | 99 | 98 |
| M | 118 | 106 | 100 | 99 | 98 | 98 |
| A | 114 | 104 | 99 | 98 | 98 | 98 |
| M | 110 | 101 | 99 | 99 | 99 | 98 |
| J | 110 | 100 | 99 | 99 | 98 | 98 |
| J | 108 | 100 | 101 | 98 | 98 | 98 |
| A | 106 | 99 | 100 | 98 | 98 | 97 |
| S | 107 | 100 | 100 | 98 | 98 | 97 |
| O | 108 | 100 | 101 | 99 | 98 | 98 |
| N | 115 | 104 | 98 | 98 | 98 | 98 |
| D | 119 | 107 | 100 | 101 | 99 | 98 |

**Fig 6. Effect of mass estimate (*Plateosaurus*) on annual energy.** The matrix reflects the effect of size and insulation for *Plateosaurus* in a hot and cold microclimate. Dark blue = >10% above target ME; light blue = 5–10% above target ME; green = +/- 5% of target ME; light orange = 5–10% below target ME; and dark orange = >10% below target ME.

reduction in metabolic heat production during some parts of the year, thus requiring less food intake. Under cold climate conditions the uninsulated *Coelophysis* with a carnivorous diet requires more than twice the target food intake to maintain an elevated, ratite like core temperatures, while a fully insulated individual is slightly heat stressed requiring less than the target food intake. This heat stress is overcome with a slight reduction in insulation or a broadening of CTR (see below).

It is notable that the absolute difference between the cold and hot climate annual food requirements decreases non-linearly as insulation increases similar to that reported by Porter [101]. There is a 6% difference in the annual food budget between hot and cold climates for the fully insulated *Coelophysis* and an 8.8% difference for *Plateosaurus* for all diets (carnivorous/herbivorous). In contrast, the difference in annual food budget under cold and hot climates for the top-only insulated and uninsulated *Coelophysis* increases to 36% and 46% respectively for the cold climates relative to warm climates. These differences in food

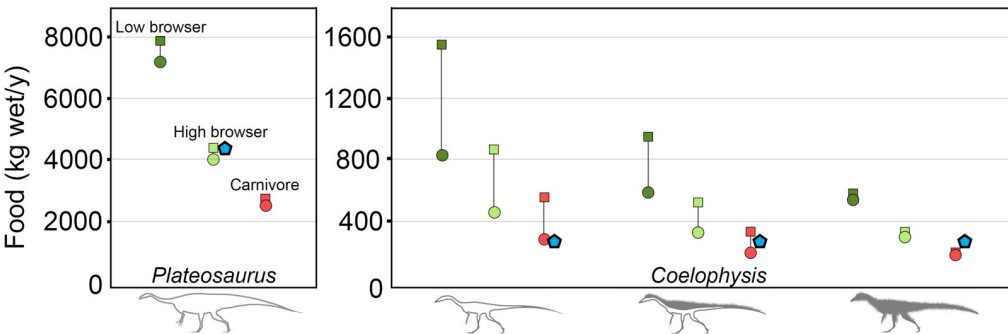

**Fig 7. Dietary variability with diet type and insulation.** The amount of food needed to maintain the specified (*target*) core body temperature throughout the year varies with diet type. Diet types: low browser herbivore (dark green); high browser herbivore (light green); and carnivore (red). Climate conditions also affect the quantity of food required to maintain core temperatures in hot (closed circles) and cold (closed squares) climates; annual target food intake in kilograms for each species is denoted by a closed blue pentagon when *Plateosaurus* = high browser and *Coelophysis* = carnivore. Data represent each species with a ratite RMR and CTR.

requirements for small dinosaurs with little to no insulation are directly related to the decrease of thermal heat flux from the body due to increased insulation for fully insulated *Coelophysis* or having a large adult body size like *Plateosaurus*.

## Metabolic chamber simulation results

Within the metabolic chamber simulation that spanned 0–50°C, *Plateosaurus* displayed a greater range of temperatures where it could remain in its active-thermoneutral zone relative to the small bodied uninsulated and top-only insulated *Coelophysis*. The fully insulated *Coelophysis* exhibited a similar breadth of thermoneutrality range as the *Plateosaurus* (Fig 8). For

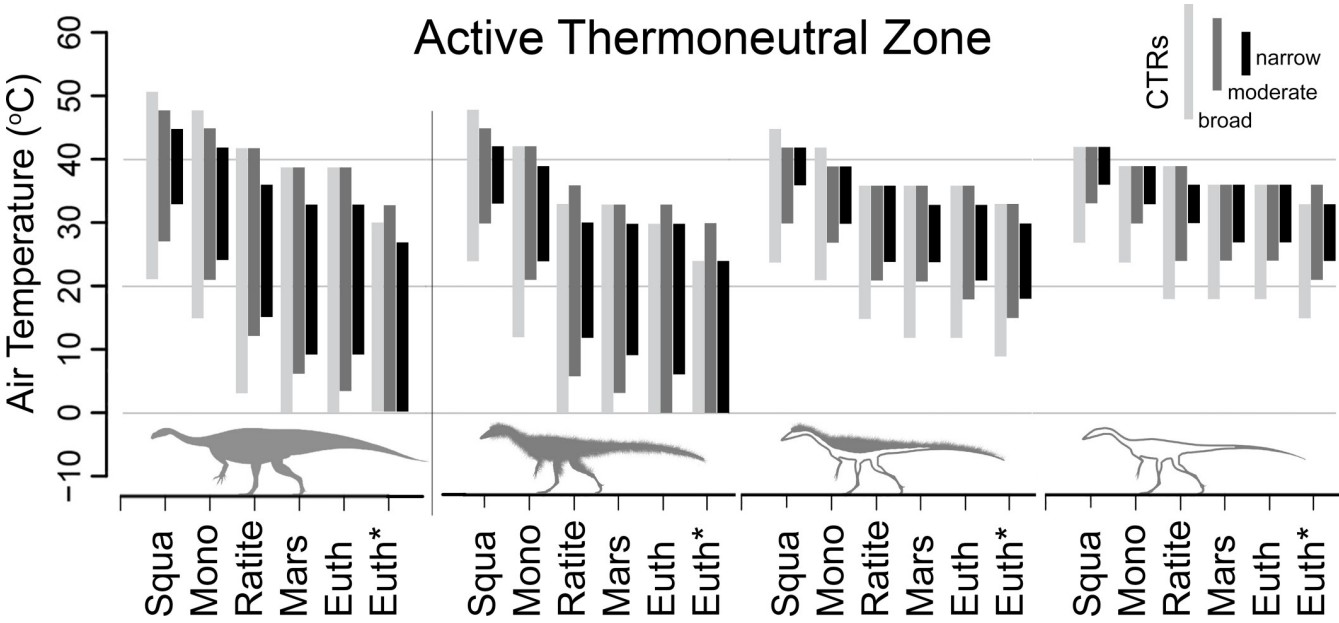

**Fig 8. Active thermoneutral zones of *Plateosaurus* and variably insulated *Coelophysis*.** Shaded areas represent the active thermoneutral zone determined from 18 metabolic chamber experiments for *Plateosaurus* and *Coelophysis* (fully insulated, top-only, and uninsulated) with RMR ranging from squamates to eutherians based on published regression equations [7,75,76]. Light gray = broad CTR (26–40°C); dark gray = moderate CTR (32–40°C); black = high CTR (36–40°C). Target $T_{core}$ = 38°C. The active thermoneutral zones for the top-only and fully insulated *Coelophysis* were calculated with the ptiloerection behavioral function enabled.

each stepwise increase in resting metabolic rate (RMR; from squamate to eutherian) two general trends were observed: 1) thermoneutral zone breadth increased and 2) the maximum and minimum thermoneutral temperature values each shifted to lower values. This trend is more apparent in the larger bodied *Plateosaurus*. For example, *Plateosaurus* with a ratite-like core temperature range (CTR) of 38±2°C can maintain thermoneutrality with a squamate grade RMR in air temperatures between 32–45°C and between 15–36°C with a ratite grade RMR. There is an 8°C increase in the absolute thermoneutral range from squamate to ratite RMR, while the maximum air temperature shifts negatively by 9°C.

Varying the amount of external insulation in the form of filamentous 'proto'-feathers made a substantial difference in thermoneutral temperature ranges. An uninsulated *Coelophysis* could maintain thermoneutrality over an 6–10°C temperature range (Fig 8). With a ratite RMR and CTR (38±2°C) the thermoneutral range of an uninsulated *Coelophysis* was 30–36°C. As dermal insulation was added the overall pattern observed was similar to that seen with increased BMR; i.e., there was a stepwise decrease in maximum and minimum thermoneutral temperature, but an overall increase in total range. The thermoneutral range relative to the uninsulated model was extended moderately 0–3°C (depending on metabolic rate) in the top-only insulated *Coelophysis*. A fully insulated *Coelophysis* had a substantial decrease in the lower end of its thermoneutral range while minimally decreasing its upper bound (12–30°C); the fully insulated *Coelophysis* more than doubled its active thermoneutral air temperature range. The net effect of insulation allows a fully insulated *Coelophysis* to maintain thermoneutrality across a much broader temperature range in colder environments compared to the non-insulated *Coelophysis*, although this is at the cost of lowering the maximum tolerable air temperature.

To test the effect of variable CTRs as well as RMRs we simulated a broad (26–40°C), moderate (32–40°C), and narrow (36–40°C) core temperature range for each of the 6 RMRs (Fig 8). The same trends were observed with the broad and moderate CTR as seen in the narrow CTR simulations, however the absolute range was greatest in the broad CTR and intermediate in the moderate CTR, and lowest in the narrow CTR discussed above. We also tested the model with four different *target* core body temperatures, 38, 35, 32 and 29°C with a narrow (±2°C) and broad(+2/-13°C) CTR for both species to compare their active thermoneutral zones under these conditions. As the target core temperature was stepped down, the overall thermoneutral range remained effectively the same, but the absolute minimum and maximum air temperature values shifted negatively ~ 2–3°C for each 3°C step down in target core temperature. The results for the four target core temperatures under ratite-like and squamate-like CTR are shown in S4 Appendix Figs 9 & 10, respectively.

### Effects of resting metabolic rate and core temperature range

Each model simulation paired different physiological combinations of resting metabolic rate (RMR; squamate, monotreme, and ratite grade metabolic rates) with a broad, moderate, or narrow core temperature range (CTR), each with a 38°C target core temperature (26–40°C, 32–40°C, and 36–40°C, respectively), under cold, moderate, and hot climates for the two dinosaur species. The results of these experiments yielded hourly outputs that were plotted as annual heatmaps for core body temperature, metabolic energy (contoured in multiples of RMR), and hours in open versus shaded conditions (see Fig 9 for explanation of heatmaps).

*Coelophysis* **(uninsulated).**   The uninsulated *Coelophysis* model results show a high degree of cold stress for all but 3 of the 27 possible RMR/CTR/microclimate combinations (Fig 10). The two best fits are the moderate and upper RMR with broad CTR in the hot microclimate. However, under all RMR/CTR combinations *Coelophysis* is cold stressed in the cold

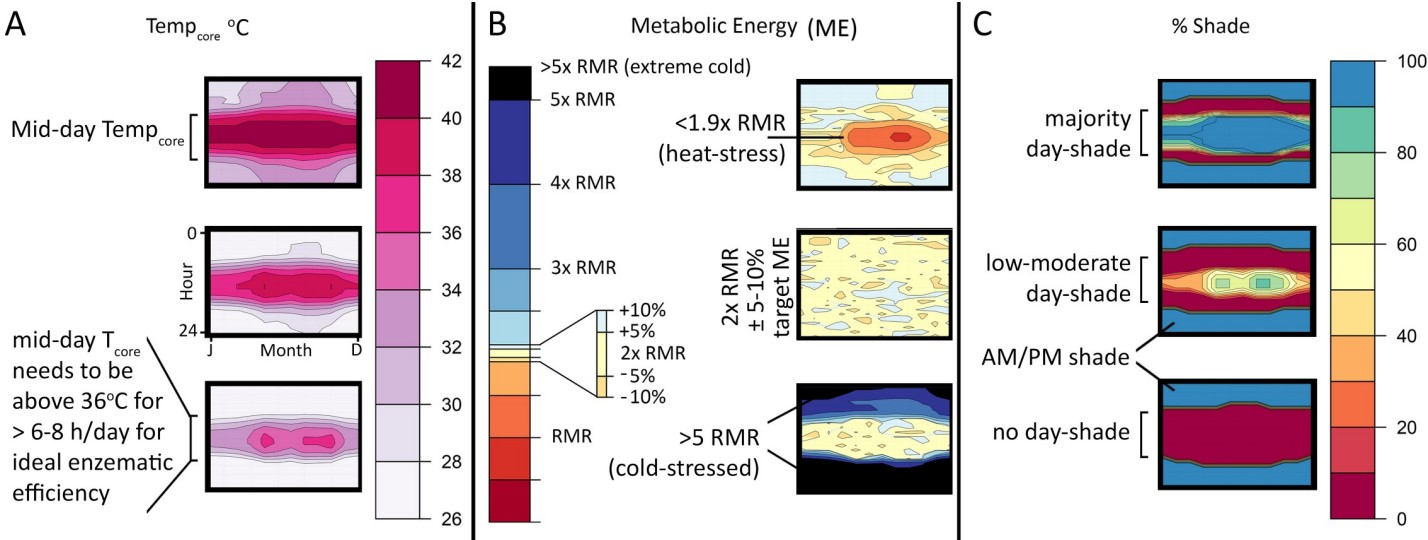

**Fig 9. Heatmaps of T_core, metabolic energy (ME), and % shade.** Heatmaps provide a quick quantitative tool for visualizing results on an hourly basis across the year. A) Top; example of narrow CTR heatmap where $36 < T_{core} < 40$ ~6–8 hrs per day. Bottom; $36 < T_{core} < 38$ ~0–3 hrs per day (e.g., cold stressed). B) Top; Metabolic energy (ME) heatmap displaying heat stress during mid-day hours, mid-year. Middle; ME heatmap displaying a reasonable range around 2x RMR. Bottom; results of a cold-stressed model with ME exceeding 5x RMR. C) Heatmaps demonstrating a high (top), moderate (middle), and low (bottom) daylight hours shade requirement.

microclimate. Even if paleotemperatures of high latitude localities were only slightly cooler (moderate) than equatorial (hot) conditions modeled herein, the uninsulated *Coelophysis* still shows signs of cold stress (i.e., T_core does not reach 35°C for more than 4 hours a day, for over half of the year; 3 months of the year never reach 35°C at all). This lends support to a requirement of some form of insulation or thermoregulatory behavior under all model conditions, leaving room for the possibility that uninsulated adult *Coelophysis* could exceed modeled temperatures in the hot microclimate.

Because many ectothermic animals have the potential to decrease their internal temperatures below the 26°C lower bound we used in the broad, squamate-grade CTR, we also modeled the uninsulated *Coelophysis* with a 10°C lower temperature bound to ensure we capture the lowest extremes of core temperature. *Coelophysis* was modeled in the hot and cold microclimate for the month of May (northern hemisphere early summer). These data were plotted along with the November (southern hemisphere early summer) temperature profile for the largest known extant predatory ectothermic terrestrial vertebrate, *Varanus komodoensis*, as a frame of reference (Fig 11). In the hot microclimate T_core for *Coelophysis* responded similarly to *V. komodoensis* during the modeled month. However, during the winter months the squamate-grade *Coelophysis* was slightly cold stressed without the ability to burrow like *V. komodoensis* (we assume *Coelophysis* does not burrow). In the cold microclimate T_core does not exceed 30°C for more than 5 months of the year demonstrating severe cold stress for the uninsulated *Coelophysis* (non-viable).

With a 10–40°C CTR the lowest ambient air temperature in the cold microclimate was above the lower (10°C) CTR threshold, thus, it was possible for the organism to thermoregulate and maintain its target ME by dropping its core temperature rather than increase its metabolic rate (see S4 Appendix Fig 11). This did not affect the daily core temperature results between 26 and 40°C which are identical as the prior broad CTR experiment above; the animal is still significantly cold-stressed.

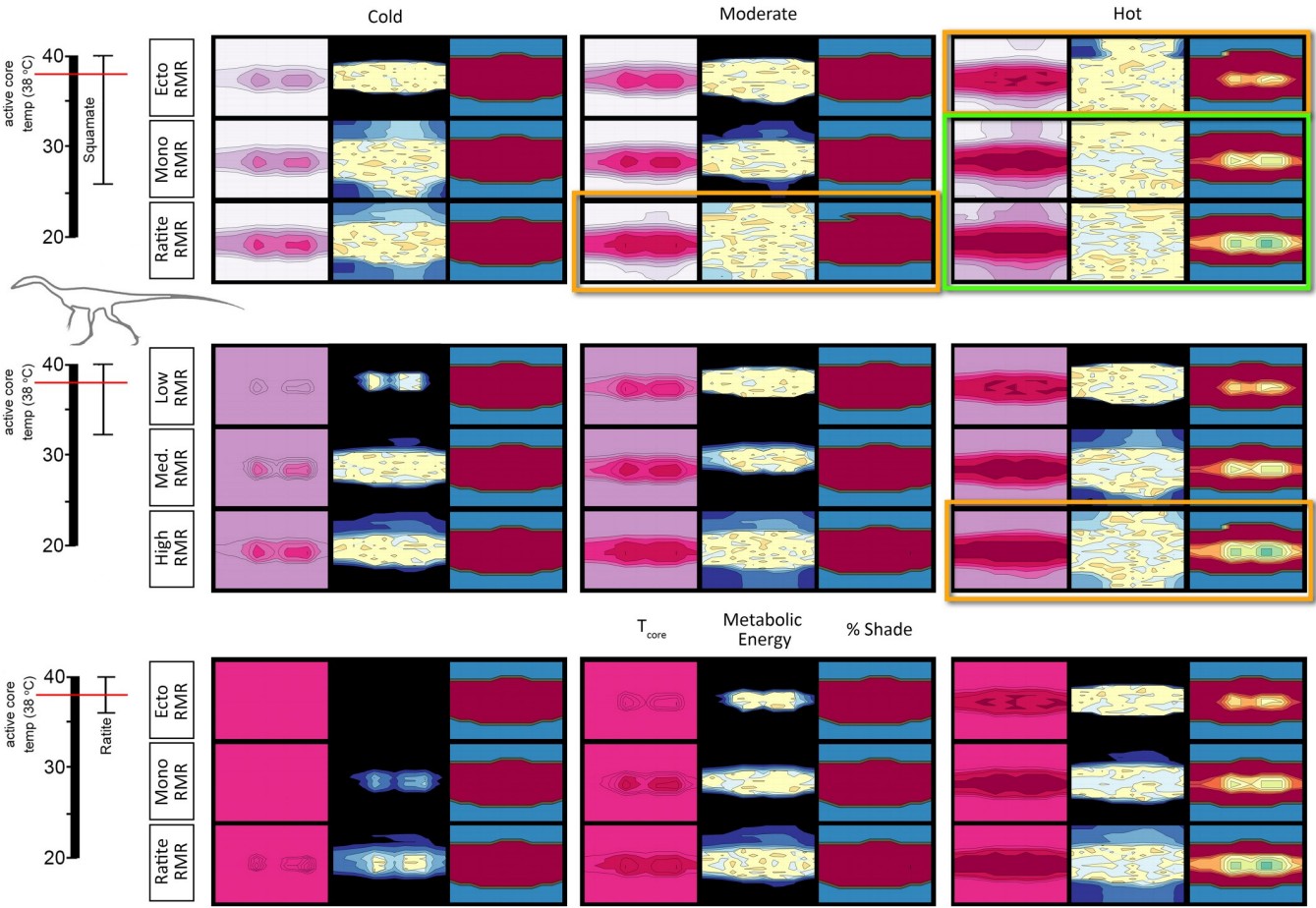

**Fig 10. T$_{core}$, ME, and %shade heatmaps for *Coelophysis* (uninsulated).** Heatmaps representing the hourly results across the modeled year the three dominant variables: microclimate (hot, moderate, and cold), RMR (low, moderate, and high), CTR (broad, moderate, and narrow) for an uninsulated *Coelophysis*. See Fig 9 for key. Two most likely scenarios for survivability are outlined in bright green, the three edge conditions are outlined in orange; all other conditions are considered to be non-viable.

*Coelophysis* (**top-only insulation**).    With the addition of insulation to the top-half of *Coelophysis* the severity of cold-stress decreased and the number of viable RMR/CTR/microclimate combinations increased to 6 of 27 (Fig 12). Under the hot microclimate with a broad CTR, all RMR conditions met ME targets and were able to maintain a core temperature above 35˚C. As the microclimate shifted to the moderate condition, the lower RMR was excluded; all RMR were excluded under the cold microclimate. As the CTR reached the moderate range, only the moderate and upper RMR were considered feasible under the hot microclimate. The narrow CTR excluded all RMR in all microclimates.

*Coelophysis* (**full insulation**).    With a fully insulated *Coelophysis* the severity of cold-stress further decreased and the number of viable RMR/CTR/microclimate combinations increased to 10 of 27; heat stress was evident in all 3 CTRs with an upper RMR under the hot microclimate (Fig 13). The fully insulated *Coelophysis* was cold stressed in the cold and moderate microclimates with a broad CTR. However, it was able to maintain its ME target and sustain a core temperature greater than 34–36˚C for at least half of dial hours with a broad CTR and: lower RMR in the hot microclimate; moderate RMR in moderate and hot microclimates; upper RMR in cold and moderate microclimates. Raising the CTR to the moderate condition

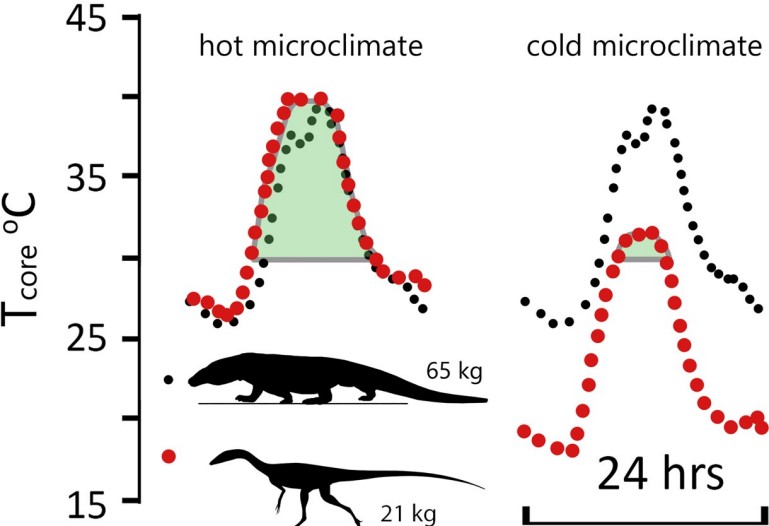

**Fig 11. Comparison of daily temperature curves for *Varanus* and *Coelophysis* (uninsulated).** Daily temperature curves for hot and cold microclimates for the 15th of May (uninsulated *Coelophysis*) and November (*Varanus komodoensis*) [40]. There is strong agreement between the low RMR and broad CTR *Coelophysis* and *V. komodoensis* in the hot microclimate. *Coelophysis* modeled in the cold microclimate was significantly cold stressed. Green shaded area represents duration of day with $T_{core} > 30°C$.

excluded all lower RMRs as well as the moderate RMR condition in the moderate microclimate. A narrow CTR resulted in the loss of the remaining moderate RMR in the hot microclimate; only the upper RMR in moderate and cold microclimates were able to meet their ME target.

**Plateosaurus.** *Plateosaurus* exhibits a similar response to that seen in the fully insulated *Coelophysis*; the number of viable RMR/CTR/microclimate combinations was 10 of 27; heat stress was evident in all 3 CTRs with a ratite RMR under the hot microclimate (Fig 14). With a lower RMR and broad CTR, *Plateosaurus* was cold stressed in the early morning hours under cold conditions and didn't exceed 30°C body temperature for more than half of the calendar year. The moderate microclimate fared only slightly better, but the ME still exceeded target by ~10%. This same physiology modeled in the hot microclimate demonstrated a core temperature of 28–30°C during morning hours and reached target core temperatures by midday.

When the CTR reached the moderate level, all lower RMR were excluded due to significant cold stress, as was the moderate RMR in the cold microclimate. The moderate RMR met its ME target in the hot microclimate, but its ME slightly exceeded its target goal in the moderate microclimate and exceeded its ME target under the cold microclimate. The final step to a narrow CTR increased the cold stress previously observed in the moderate and cold microclimate with a moderate RMR, the model slightly exceeded its ME target under the cold microclimate with an upper RMR. The model met its ME target under the moderate microclimate with an upper RMR.

## Microclimate wind effects

Because the wind was the second strongest main effect in our yates analysis (see S4 Appendix) we further explored this effect using *Coelophysis* and *Plateosaurus* with an upper RMR and narrow CTR. For *Coelophysis*, the magnitude of wind effects varies substantially depending on the degree of insulation, ptiloerection, and climate (Fig 15). Daily variation in wind speeds from 0.1 to 2.0 m/s affects total annual energy requirements from approximately 2000 (hot

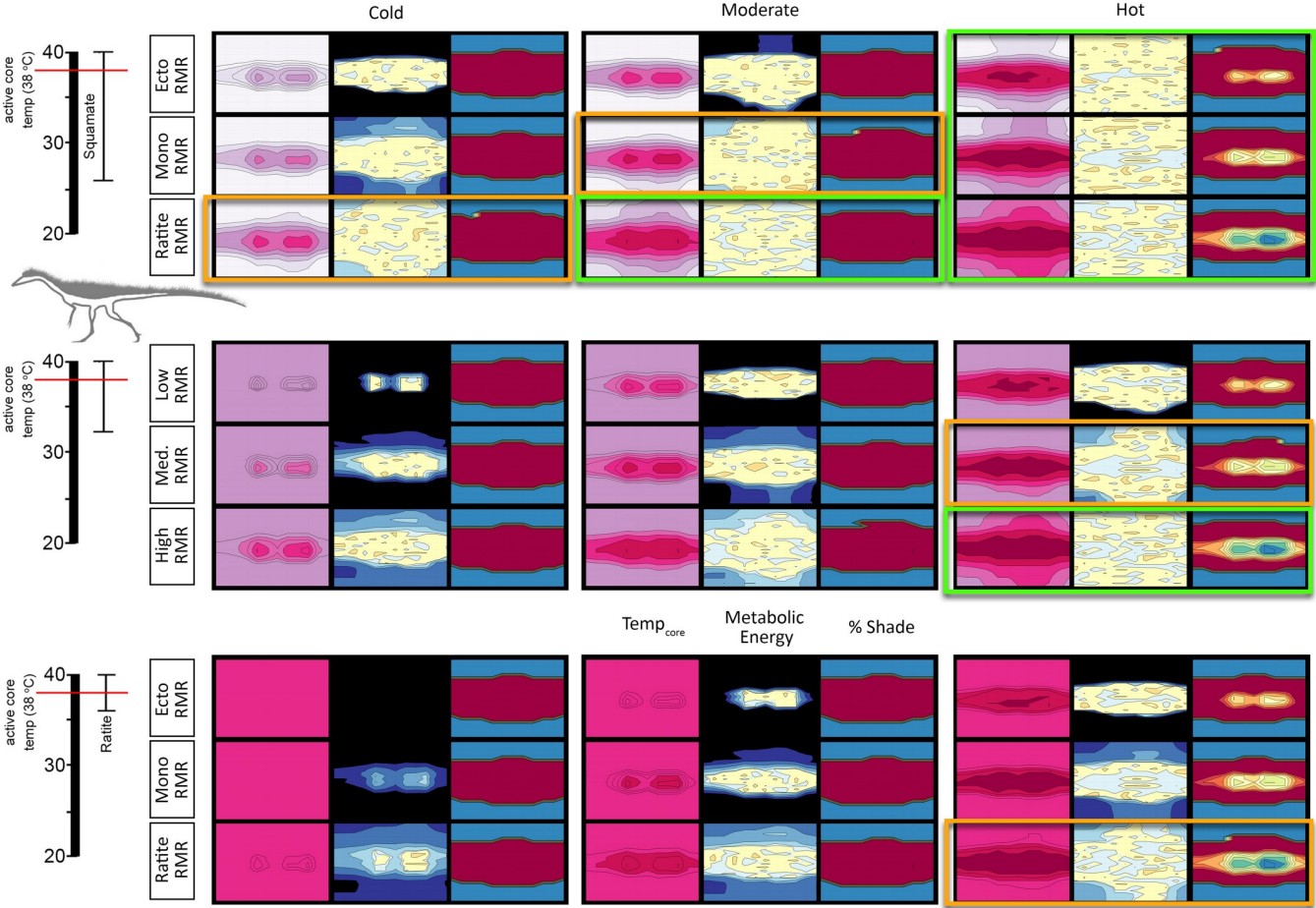

**Fig 12. T_core, ME, and %shade heatmaps for *Coelophysis* (top-only insulated).** Heatmaps representing the hourly results across the modeled year the three dominant variables: microclimate (hot, moderate, and cold), RMR (low, moderate, and high), CTR (broad, moderate, and narrow) for a top-only insulated *Coelophysis*. See Fig 9 for key. Five most likely scenarios for survivability are outlined in bright green, the four edge conditions are outlined in orange; all other conditions are considered to be non-viable.

microclimate) to 3400 MJ/y (cold microclimate) without insulation down to approximately 1400 (hot microclimate) to 1800 MJ/y (cold microclimate) when fully insulated *without* ptiloerection (30 mm insulation depth); the fully insulated (with ptiloerection enabled) was ~1500 MJ/y. Ptiloerection was not activated until the model required >2x RMR to maintain *target* core temperatures.

We tested 5 insulatory conditions for each climate: 1) no insulation, 2) 15 mm depth top half only (only the top half of the animal had insulation), 3) 30 mm depth top half only with, 4) 15 mm depth fully insulated, and 5) 30 mm depth fully insulated. Fully insulated animals only engaged ptiloerection in the coldest microclimate. When the feather depth of the fully insulated animal decreased from 30 to 15 mm its energetic response was similar to that of the 30 mm top-only insulation; thus decreased insulation depth is equivalent to greater depth with only top surfaces insulated.

Wind did not have as large of an impact on the modeled *Plateosaurus*, although wind was the second strongest effect observed in the Yates analyses. *Plateosaurus* was able to maintain its target core temperature under the moderate and hot microclimates for average and high speed winds. In low speed winds moderate and cold microclimates were at the lower *target*

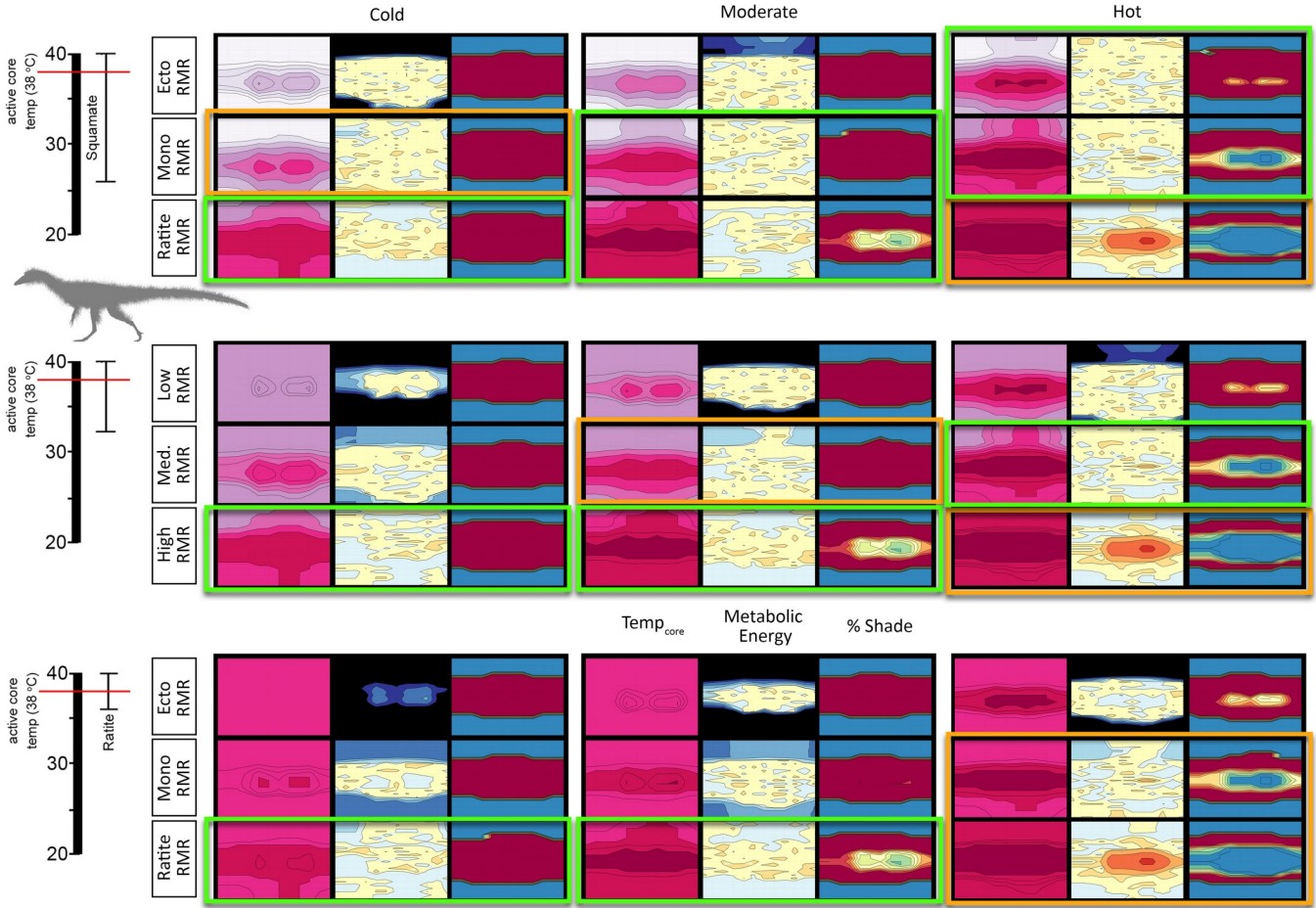

**Fig 13. T$_{core}$, ME, and %shade heatmaps for *Coelophysis* (fully insulated).** Heatmaps representing the hourly results across the modeled year the three dominant variables: microclimate (hot, moderate, and cold), RMR (low, moderate, and high), CTR (broad, moderate, and narrow) for a fully insulated *Coelophysis*. See Fig 9 for key. Tenmost likely scenarios for survivability are outlined in bright green, the six edge conditions are outlined in orange; all other conditions are considered to be non-viable.

boundary (-5% of 2x RMR), while the hot microclimate caused notable heat stress (-10% of 2x RMR; Fig 16). This stresses the importance of behavior for the model to seek shelter from or take advantage of higher wind conditions for thermoregulation.

## Discussion

Mechanistic physiological modeling of extant organisms accurately predicts environmental range with high fidelity [19,20,31,32,69]. This has been leveraged to generate hypotheses of how organisms respond to habitat expansion, contraction, and altered geographic ranges associated with changing climate on local and global scales [12,18,40,102]. Niche Mapper in particular has demonstrated the ability to predict metabolic expenditure as a function of environmental conditions for a broad sample of vertebrates in microclimates ranging from arctic to tropical [13,17,20,32,34]. Our efforts have focused on extending Niche Mapper to generate and test biophysical hypotheses against known paleobiogeographic distributions, phylogenetic position, and life histories for two extinct animals in deep time. While we lack detailed empirical profiles for the physiology of extinct animals, we can explore different

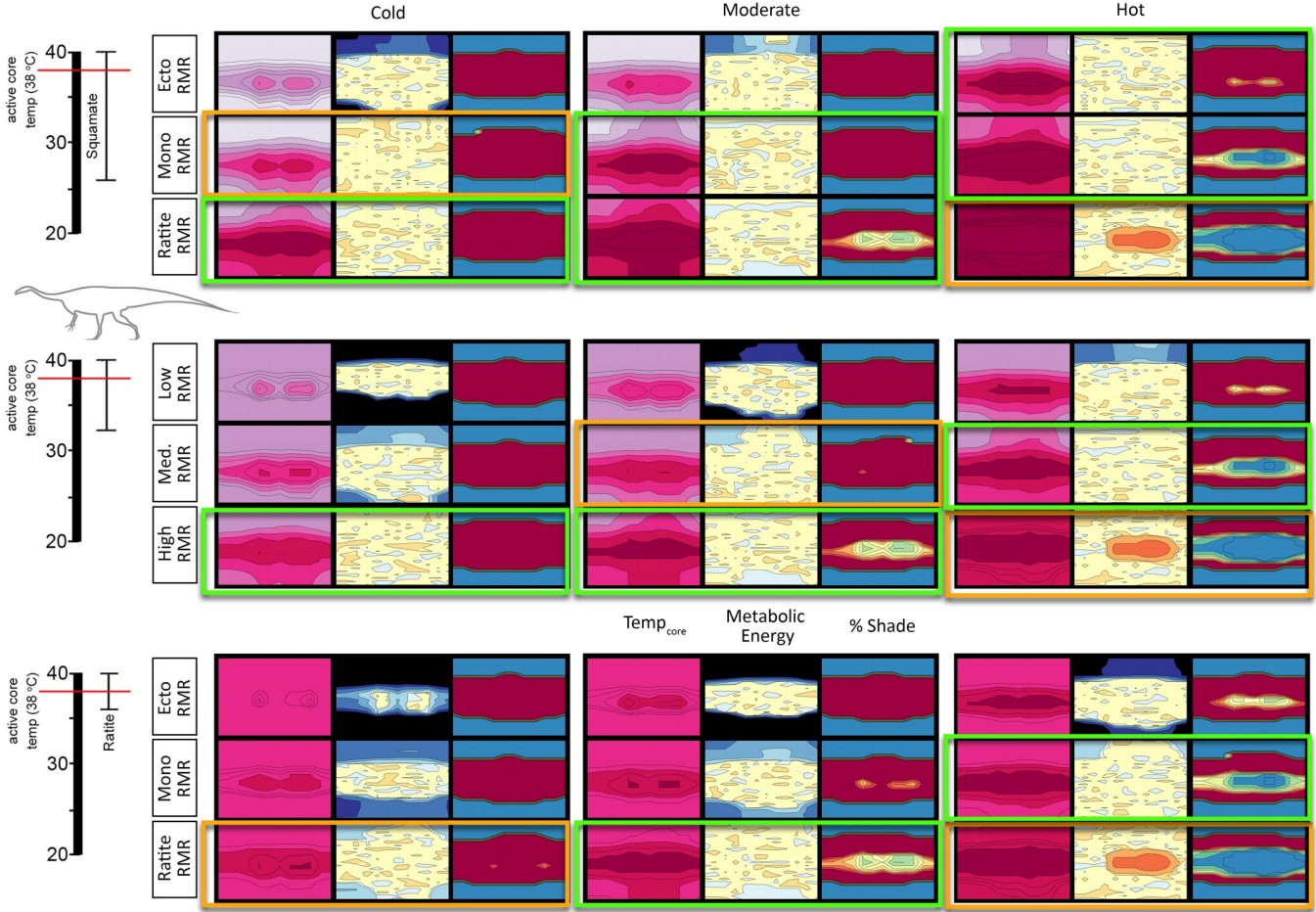

**Fig 14. T<sub>core</sub>, ME, and %shade heatmaps for *Plateosaurus*.** Heatmaps representing the hourly results across the modeled year the three dominant variables: microclimate (hot, moderate, and cold), RMR (low, moderate, and high), CTR (broad, moderate, and narrow) for a *Plateosaurus*. See Fig 9 for key. Ten most likely scenarios for survivability are outlined in bright green, the six edge conditions are outlined in orange; all other conditions are considered to be non-viable.

combinations of morphological and physiological characteristics to determine their effect on energetics, behavior, and animal distributions.

We utilized a variety of stable isotope and geochemical systems to infer mean annual temperature, mean annual precipitation, atmospheric $O_2$ and $CO_2$, and relative humidity in the Late Triassic of Western North America [55,103–105]. These and other sedimentary paleoenvironmental proxies [e.g., 52] were employed to refine global climate model data used to construct our microclimate models. We also decoupled our organismal models from specific microclimate models using Niche Mapper's virtual metabolic chamber function to determine an active thermoneutral temperature range for modeled taxa. Determining the overlap in thermoneutral zones of organisms and mapping them against their paleobiogeographic distributions [e.g., 15,23] provides an independent test of plausible paleophysiologies.

## Deep-time model uncertainty

In our deep-time implementation of Niche Mapper, we have been forced to make a series of assumptions. We assume the distribution of fossils are evidence of the presence of a viable population of organisms in that time and place. We further assume that the range of extant

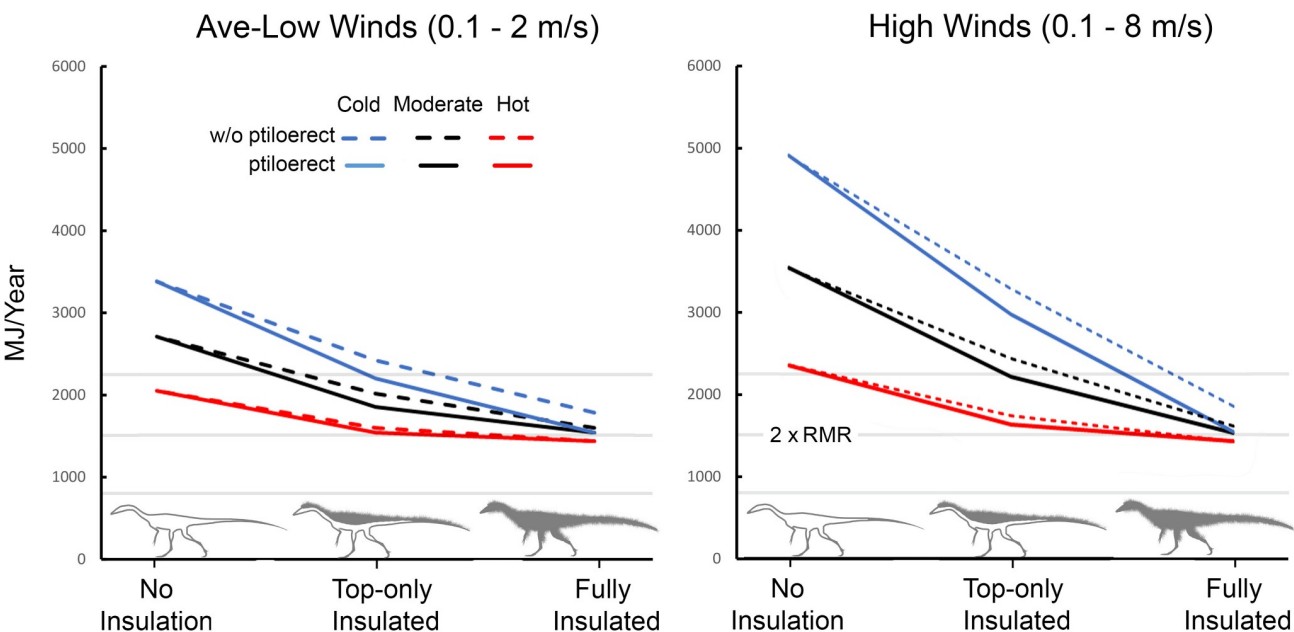

**Fig 15. Energetic cost of wind exposure for *Coelophysis*.** As temperature increases (blue, black, and red lines, respectively) ptiloerection was less beneficial with increased insulation volume. (e.g., fully insulated *Coelophysis* does not significantly benefit by implementing ptiloerection at hot temperatures but the presence of feathers broadens its active thermoneutral zone). The three light gray horizontal lines represent, from bottom to top, resting, twice resting (e.g., ME target), and three times resting metabolic rate to indicate the likely range of activity levels for the size, shape, and degree of insulation for *Coelophysis*.

tetrapod physiology (e.g. newts to birds) bound possible physiologies for these Triassic organisms. At this time we find no convincing evidence that a physiology outside of observed modern bounds existed for these organisms. We also assume paleoclimate proxies provide reasonably accurate reconstructions of local environments and that parameters which do not have reliable proxies (i.e. wind speed, cloud cover) can be bound by modern environments with similar temperature-precipitation regimes. We further assume that conditions not currently possible were likewise not possible in the Triassic (e.g. 100 km/hr constant winds, annually). Some uncertainty in model interpretation may be created by these assumptions, and additional uncertainty may derive from the absence of fossil evidence required to constrain

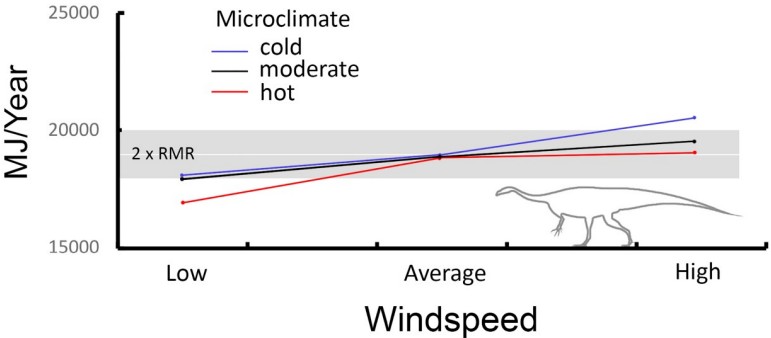

**Fig 16. Energetic costs of wind exposure for *Plateosaurus*.** Under low wind speeds *Plateosaurus* is moderately to notably heat stressed (cold/moderate and hot, respectively). *Target* ME is maintained in all microclimates for average winds, and in the hot microclimate with high winds. *Plateosaurus* becomes cold stressed with high winds in the cold microclimate. Red line = hot microclimate, blue line = cold microclimate, black line = moderate microclimate.

behavioral thermoregulation such as the use of burrows, shelters, or torpor as a means of altering heat transfer to and from the environment.

Suites of inferences are testable. If a modeled combination of physiology, shape, behavior, insulation, and climate do not produce a viable organism in a location where there are fossils, then clearly the hypothesized input parameters are flawed. For instance, if we were to find skin impressions of multiple *Coelophysis* without insulating integument at high latitudes, but model results suggest uninsulated individuals were not viable it might be necessary to add adipose, enable torpor, allow burrowing for behavioral thermoregulation, or reassess the inferred microclimate to allow for the maintenance of a viable population. If remains of *Plateosaurus* (or similar sized prosauropods) are found in areas we have predicted thermal exclusion, then our current hypothesis would need to be rejected and the model would need to incorporate these new constraints to generate new testable hypotheses. It is also possible for the model to generate several equally parsimonious solutions. In some situations sheltering (i.e., burrows) or elevating metabolic rate may provide equally viable solutions. Moreover, our modeled organisms are adult forms of specific clades and are not representative of the ecosystem as a whole. Nor do we assume that juveniles from hatchling to subadult will be modeled with similar parameters as each other, or their adult counterpart—each ontogenetic stage may have different combinations of physiology, shape, behavior, insulation and niche availability. It is beyond the scope of this study to model each stage of ontogeny. To further explore and test the viability of our models future work will need to generate ontogenetic series for each species, and contemporaneous taxa at high latitudes such as small archosaurs, amphibians, and lepidosaurs known to co-occur with Plateosaurus and Coelophysis. This ability to incorporate new data allows for future refinement of the model and for new testable hypotheses to be generated, akin to the process of generating and testing phylogenetic hypotheses.

## Plateosaurus

Our results demonstrate that an adult *Plateosaurus* could have maintained its target metabolic energy (ME) in hot environments with either a squamate-like core temperature range (CTR) and resting metabolic range (RMR), or with a monotreme-like RMR at moderate to narrow CTR. A shift from hot to moderate or colder environments, however, required at minimum a ratite-like RMR with a moderate CTR. Modeling *Plateosaurus* with a ratite-like narrow CTR and upper RMR resulted in heat stress in hot environments, full viability in moderate environmental temperatures, and slight cold stress in our coldest environments (Fig 17).

Modeling *Plateosaurus* with a squamate RMR was non-viable for all moderate and cold microclimates regardless of CTR. The greater viability of *Plateosaurus* with elevated RMRs in Late Triassic environments is consistent with isotopic estimates derived from dinosaur teeth and eggs which suggests an elevated core temperature between 36–38°C for the sauropod lineage [106–110]. Additionally, a squamate-like broad core temperature would be near the lower limits of enzymatic efficiency (regardless of ontogeny) seen in large extant herbivores. This would translate to an inhibition of rapid growth, counter to rates of growth reported from *Plateosaurus* bone histology [111,112].

The temperate paleobiogeography of *Plateosaurus* and other Triassic sauropodomorphs precludes squamate-level RMRs and CTRs. If *Plateosaurus* was maintaining a narrow internal core temperature range then a ratite-like RMR and narrower CTR would be required. Physiological acclimatization including seasonally variable metabolic rates, variable fat stores, or changes in thermal conductivity to the ground could potentially facilitate this, as these mechanisms do in extant endothermic animals [e.g., 113,114]. We conclude that *Plateosaurus* most

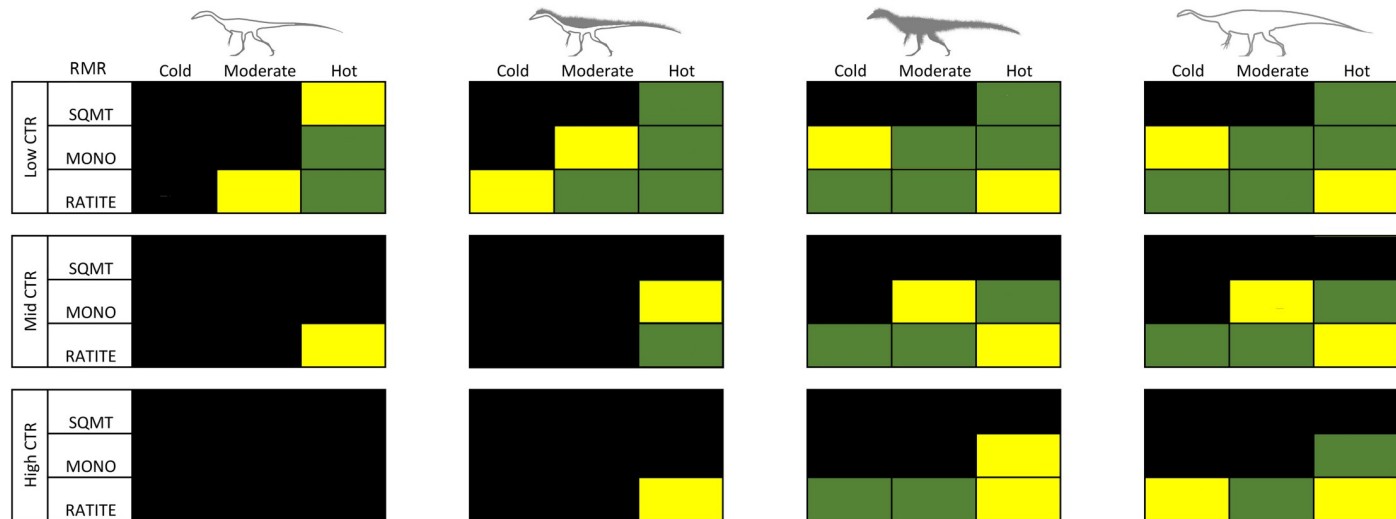

**Fig 17. Summary of viable, conditional tolerance, and non-viable results.** The matrix provides a summary of viable combinations of resting metabolic rate (RMR) and core temperature range (CTR) within cold, moderate, and hot microclimates. Green = viable; black = non-viable; yellow = conditional tolerance (e.g., a possible but extreme endmember of viability).

likely had a moderate core temperature range coupled with an elevated ratite-like resting metabolic range.

As previously mentioned we only modeled an adult *Plateosaurus*. A viable population would require survival of all ontogenetic stages through reproductive age. Preliminary simulations of isometrically scaled 'juvenile' (10-100kg) *Plateosaurus* suggest they would require either an RMR elevated above ratite grade (between marsupial and eutherian grade for a 10 kg individual, decreasing with mass) and a moderate CTR (i.e., no defence of stable core temperature), insulation (feathers or subdermal adipose), shelter (burrow or nest) or some combination of these to survive in the cold microclimate during the coldest 4 months of the year. The 10 and 20 kg 'juveniles' were viable with only the addition of sheltering behavior in the moderate microclimate. Modeling allometrically smaller individuals coupled to growth rates [111,112] is a necessary next step. It is known that juvenile (nestling) sauropodomorphs such as Mussaurus were not only allometrically different than adults (e.g., proportionally taller skulls with short snout and larger eyes, and shorter tails and necks), they were functionally different as obligate quadrupeds that only later in ontogeny gained a bipedal stance [115, 116]. It is likely that there are significant differences that will need to be accounted for when modeling juvenile vs. adult forms. Although modelling an entire ontogenetic series is beyond the scope of the current study, it's a logical step for future studies to investigate.

## Coelophysis

When *Coelophysis* is modeled without insulation while possessing either a narrow or moderate CTR it experiences excessive cold stress in all environments regardless of RMR. Non-insulated *Coelophysis* modeled with a broad CTR is able to meet its target thermoregulation in hot environments with either a monotreme or ratite-like RMR, but remains non-viable due to cold stress under all other conditions. When *Coelophysis* is modeled with a squamate-like RMR and CTR its daily $T_{core}$ profile is similar to an adult komodo dragon, each under early summer conditions (see results above). However, the uninsulated *Coelophysis* exhibits cold stress in winter months. Cold stress can be alleviated by decreasing the lower CTR bound by 2°C, but this does

not alter peak temperatures, which remain below target core temperatures with a squamate RMR. Uninsulated *Coelophysis* remains cold stressed in the moderate and cold environments as a squamate. In short, a non-insulated *Coelophysis* would be viable only in hot environments with a broad CTR, and a moderate to high RMR.

Dorsal epidermal insulation increases the capacity of *Coelophysis* to maintain its daily target ME in moderate environments with a broad CTR and ratite-like RMR. All RMRs were viable in the hot microclimate with a broad CTR. The addition of dorsal insulation made a moderate CTR viable, but only with a ratite-like RMR within hot environments. Half-insulated *Coelophysis* was non-viable with a ratite-like CTR in all environments and in all cold environments regardless of CTR.

The fully insulated *Coelophysis* was viable in a broader range of temperatures. A squamate-like RMR is required in the hot environment with a broad CTR for a fully insulated *Coelophysis*. A monotreme-like RMR is non-viable due to cold stress in colder environments regardless of CTR. *Coelophysis* with a monotreme-like RMR is viable in hot (moderate to broad CTR) and moderate (broad CTR only) microclimates. Cold and moderate environments are accessible to fully insulated *Coelophysis* with moderate to narrow CTRs with a ratite-like RMR, although heat stress occurs during peak summer temperatures in the hot microclimate.

Isotope paleothermometry of theropod teeth and eggshell indicates elevated RMRs and core temperature ranges above the levels of extant squamates [107,109,117]. Eagle and others [107] performed a clumped-isotope analysis on oviraptorosaur eggs and concluded the egg-layer had an average core temperature of 31.9 ±2.9˚C. The studied oviraptorosaurs had a mass broadly similar to *Coelophysis*, though they were more deeply nested within Coelurosauria; the analyzed specimens were found in deposits at a high paleolatitude (> 45˚N) during time of deposition [118]. An average core temperature much lower than 32˚C would likely inhibit metabolic efficiencies necessary for elevated growth rates reported for *Coelophysis* [119,120].

Lowering the target core temperature from 38˚C to 32˚C changes viable combinations of RMR, CTR, and insulation for *Coelophysis*. Squamate-like RMR and CTR are viable within hot environment models only; under both moderate and cold microclimates squamate-like uninsulated *Coelophysis* experiences extreme cold stress (with $T_{core}$ rarely exceeding 30˚C for more than a few hours a day, S4 Appendix Fig 12 & Table 1). The half-insulated *Coelophysis* with a ratite-like RMR and CTR was viable in moderate to hot environments. Fully insulated *Coelophysis* with a ratite-like RMR was viable in cold and moderate environments but is still heat stressed in the hot microclimate. Fully- to half-insulated *Coelophysis* with ratite-like RMR are viable in all environments with both elevated (38˚C) and lowered (32˚C) *target* core temperatures (S4 Appendix Table 1). These results support an elevated RMR for *Coelophysis* with a moderate to narrow CTR.

## Integrating model results with the fossil record

Body fossils of coelophysoids are known from equatorial through temperate paleolatitudes, while larger bodied plateosaurid body fossils are found in subtropical and temperate climates, but are absent from equatorial paleolatitudes (Fig 18). Given that the paleogeographic range of the two modeled species extends to temperate latitudes we substitute our cold microclimate model as a conservative surrogate for temperatures at subtropical to temperate latitudes [see 55,59,60]. Even if temperatures at higher latitudes were not as extreme as our modeled cold microclimate, they are certainly bound by our moderate microclimate. *Plateosaurus* modeled at 45˚N paleolatitude demonstrate that air temperature has a much greater effect than change in insolation (i.e. solar energy input), supporting the use of our low latitude cold (or moderate) microclimate as an analog for temperate latitudes.

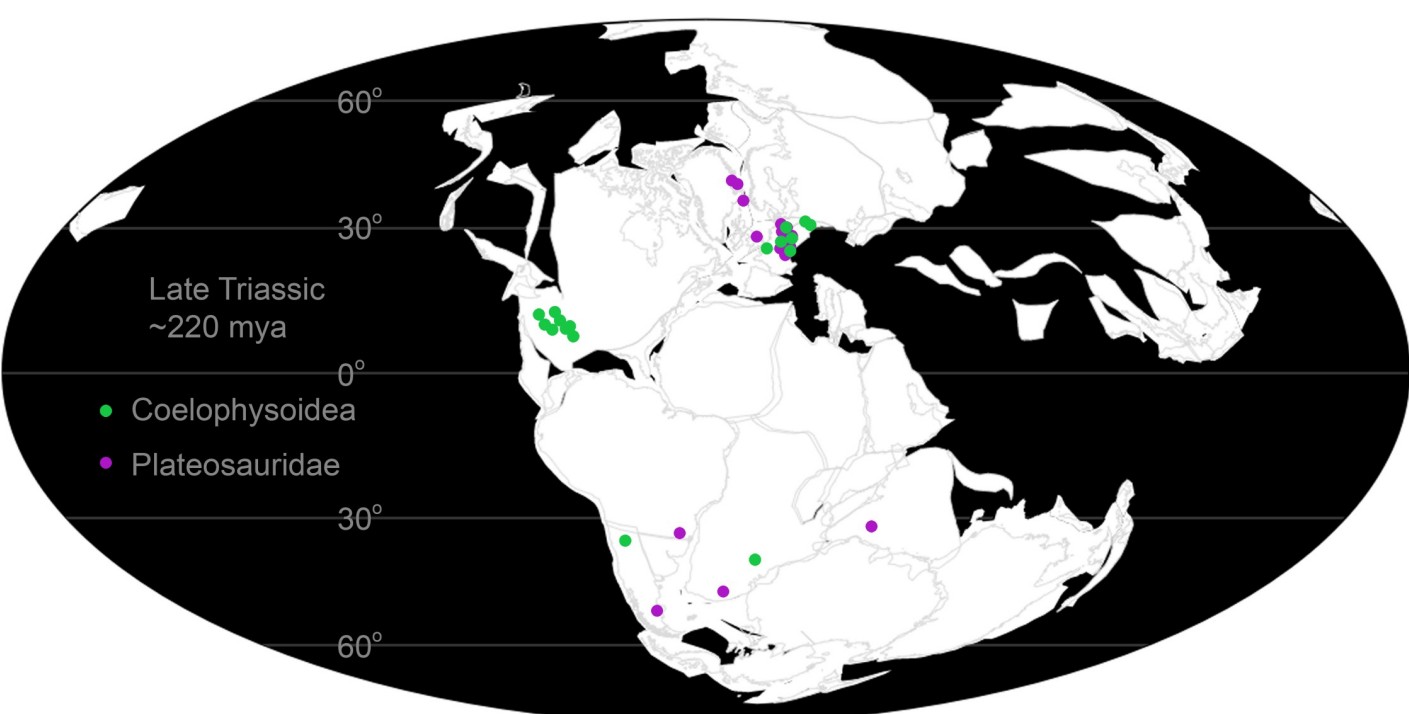

**Fig 18. Paleogeographic distribution of body fossils for members of Coelophysoidea and Plateosauridae.** Note the absence of Plateosauridae at tropical latitudes. Data from paleobiodb.org.

Skeletons of large bodied *Plateosaurus* and other large Late Triassic prosauropods (e.g., *Antetonitrus* (Yates and Kitching 2003), *Unaysaurus* (Leal and others 2004), and *Efraasia* (von Huene 1908)) have not been found in a paleogeographic gap extending through tropical latitudes. Conversely, large bodied dinosaurs are well known from cooler subtropical to temperate latitudes, consistent with our results for *Plateosaurus* in moderate to cooler environments.

Although no prosauropod body fossils have been found, the Late Triassic vertebrate track record of North America contains traces that have been attributed to medium-sized prosauropods (i.e., *Evazoum* (Nicosia and Loi 2003) formerly *Pseudotetrasauropus* (Ellenberger 1965), see [121]). The trackmakers would have been similar in size to Early Jurassic skeletons of the ~100 kg *Seitaad* (Sertich and Loewen 2010) and *Sarahsaurus* (Rowe and others 2011). There remains some doubt that the trackmakers are actually dinosaurian [121]. However, tracksites from latest Triassic-Early Jurassic sites in northeastern New Mexico have been more confidently attributed to a larger sauropod dinosaur [121]. Notably, the locations of all described prosauropod/sauropod trackways are near the edges of local highlands on the Late Triassic-Early Jurassic landscape, where they would have had access to more appropriate microclimates (Fig 19).

Whiteside and others [28] argued for climate driven environmental and vegetative instability as a dominant factor for the exclusion of large dinosaurs from the Late Triassic tropics. While we cannot rule that out, our results suggest heat stress alone would have been a significant barrier for *Plateosaurus*-sized dinosaurs. We suggest environmental temperatures limited large bodied prosauropods from greater appearance in tropical latitudes during the Late Triassic. Prosauropods may have been present in cooler environments such as forested areas and higher elevations, but these depositional environments are less conducive to fossil preservation compared to the hotter environments encountered in lowland floodplains of the Chinle

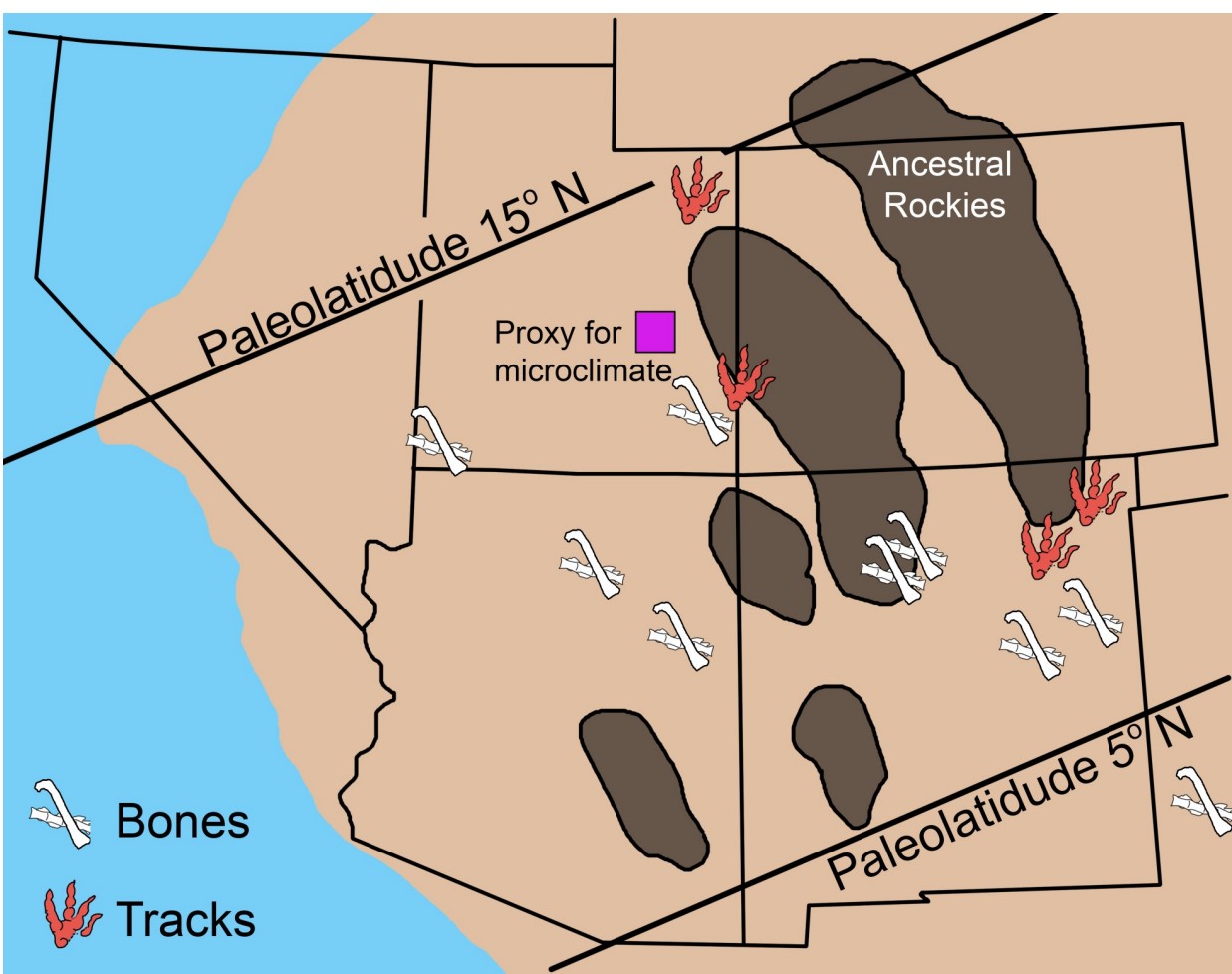

**Fig 19. Track locations attributed to prosauropods and bones of *Coelophysis* in the late Triassic of the western USA.** Purple square = field localities from which proxy microclimate data used in this study were previously published [52]. Bones = localities with known coelophysoid body fossils.

Formation [122,123]. Utilizing dense vegetative cover or higher elevations could explain the absence of body fossils in Chinle floodplain deposits but allow for the rare occurrence of track-ways attributed to prosauropod-like track makers surrounding elevated cores of the ancestral rockies (Fig 19).

It should also be noted that there are no small bodied prosauropod species known from the tropics, in contrast to several taxa known from temperate latitudes (i.e., *Thecodontosaurus* (Morris 1843) and *Pantydraco* (Galton and others 2007). Our preliminary results from isometrically scaled 'juvenile' *Plateosaurus* demonstrate 10 to 100 kg individuals are viable in the hot microclimate under the same physiological parameters (among others) as the viable adult form modeled in the cold microclimate. This is consistent with the presence of Late Triassic prosauropod-like tracks and ~100 kg Early Jurassic body fossils. Although we maintain that heat stress is a limiting factor to large bodied prosauropods in the tropics it is possible other factors such as insurmountable paleogeographic or environmental barriers, or preservational biases account for the paucity of smaller morphs within the Chinle depocenter.

Large size alone is not a limiting factor for the Late Triassic Chinle paleoecosystem. Several large bodied vertebrates inhabited this region, including the ~1000 kg dicynodont *Placerias*

(Lucas 1904) and the ~2000 kg phytosaur *Rutiodon* (Emmons 1856). Both of these taxa are capable of terrestrial locomotion but are thought to exhibit hippo-like and crocodile-like aquatic behavior and niche occupation respectively [124–126]. Spending time in water enhances heat dissipation and allows for viable large size existence in hot equatorial climates [e.g., 40]. There is no anatomical evidence to suggest prosauropods like *Plateosaurus* exhibit similar behaviors, but it cannot be excluded as a potential thermoregulatory behavior.

Other behaviors such as burrowing or hibernation were excluded in our study as they are unlikely means of temperature regulation for adult Coelophysis and Plateosaurus. However, several smaller dinosaurs from temperate latitudes later in the Mesozoic are known or suspected burrowers [127,128]. It is not outside the realm of possibility that juvenile Plateosaurus (or other small prosauropods such as the 15 kg Thecodontosaurus) would have leveraged the benefits of burrowing. Fossorial behavior would allow for exploitation of more stable microhabitats in environments with higher variance in daily or annual air temperature. It is well known that small squamates, as well as the large Varanus and crocodylian Alligator use burrowing behaviors to thermoregulate, surviving extreme weather events and even wildfires [40,129,130].

A potential discrepancy with our results is the presence of small-medium bodied taxa contemporaneous with Plateosaurus such as Thecodontosaurus, early turtles (i.e, Proganochelys), and gracile crocodylomorphs (i.e., Terrestrisuchus) which would likely model as cold stressed during the winter nights if provided similar behavior parameters without the benefit of dermal insulation or greater size. These high latitude species almost certainly employed alternate thermoregulatory strategies such as burrowing, aestivation, or hibernation, much like their modern descendants. It may be that denning (burrowing) behavior was an ancestral state for crocodyliformes [131]. It should be noted too, that while Terrestrisuchus and its relatives may not have had epidermal insulation (currently phylogenetically constrained to Ornithodira), either croc-line archosaurs or archosaurs in general may have had elevated resting metabolic rates based on respiratory anatomy [132], upright locomotion [133] and inferred growth rates [134]. Investigating the impact of those metabolic and behavioral inferences on non-dinosaurian archosaurs is an exciting avenue of potential research, for ourselves and others who adopt mechanistic modeling techniques.

## Insulation in Triassic theropods

Late Triassic theropod body and trace fossils are well known from tropical latitudes in contrast to the skeletal and ichnological fossil record of prosauropods. The 21 kg *Coelophysis* is best represented in the body fossil record of the Chinle Formation (Fig 19), and other coelophysoids are known globally at higher latitudes (Fig 18). The basal saurischian *Chindesaurus* (Long and Murray 1995)—which at the least filled a theropod-like ecological role—along with basal theropods *Daemonosaurus* (Sues and others 2011) and *Tawa* (Nesbit and others 2011) as well as the neotheropod *Camposaurus* (Hunt and others 1998) are all known from body fossils within the Chinle Formation. There are also abundant trackways attributable to theropod dinosaurs throughout the region.

According to several independent methods of mass estimation [e.g. 94,100] adult theropod taxa from this formation were near 20 kilograms. Body fossils of coelophysoids such as *Zupaysaurus* (Arcucci and Coria 2003), *Liliensternus* (Welles 1984), *Procompsognathus* (Fraas 1913), and *Coelophysis rhodesiensis* (Raath 1969) are known from subtropical to temperate paleolatitudes. *C. rhodesiensis* was discovered in temperate southern paleolatitudes and is similar in size to Chinle coelophysoids. *Zupaysaurus* and *Liliensternus* would have been around 100 kg heavier than *Coelophysis*.

*Coelophysis* and other primitive theropods had a bipedal upright stance and a narrow, laterally compressed body that reduces solar cross-section when the sun is overhead while maximizing surface to volume ratio, enhancing radiative cooling relative to more round-bodied taxa [11]. When the sun was low in the sky laterally compressed animals have greater behavioral flexibility in adjusting their solar radiation cross-section, either by facing towards the sun to minimize their cross-section or by orienting themselves perpendicular to the sun, making their solar absorption equivalent to more rotund organisms. In both cases the change produces a reduction in solar absorption during peak thermal stress, allowing for higher metabolic rates during the day [101]. Our model results show this bauplan is appropriate in warm environments, but without insulation individuals would have been at a distinct disadvantage in cooler climates.

A major difference between our *Plateosaurus* and *Coelophysis* models is the inclusion of the aforementioned states of dermal insulation in the form of primitive filamentous structures for *Coelophysis*. Filamentous and/or quill-like structures are known in a wide range of coelurosaurian theropods, basal ornithischians such as *Tianyulong* (Zheng and others 2009) and *Kulindadromeus* (Godefroit and others 2014), as well as the more derived ornithischian *Psittacosaurus* (Osborn 1923). Comparisons of pterosaurian pycnofibers and dinosaurian quill structures has led to the suggestion that the epidermal insulatory structures may be the primitive conditions for Ornithodira (i.e the most recent common ancestor of pterosaurs, dinosaurs and all their descendants; [135]). This hypothesis has been questioned by character optimization analyses [136], though they acknowledge that Early Jurassic theropod resting imprints with epidermal structures [137] makes proto-feathered Triassic theropods plausible.

In the absence of definitive skin impressions, our model can add to the discussion on the need for insulatory structures in *Coelophysis* [138–140]. We have not added such structures to Plateosaurus, as there are no prosauropod or sauropod body (or trace) fossils that demonstrate insulatory epidermal structures. If insulatory coverings were primitive to Dinosauria, it is likely they were lost as sauropodomorphs increased in size—increased mass alone can expand tolerance of cooler temperatures and stabilize internal temperature variation, but not without its own energetic costs. There is also the possibility that insulation is present in the hatchlings of some larger species and is lost during ontogeny, though scales are known in some embryonic titanosaur sauropods [141].

Skeletal remains of *C. bauri* (our model) are well known from Chinle deposits best represented by our hot microclimates while similarly sized *C. rhodesiensis* is well known from the Elliot Formation (Zimbabwe) which was deposited in a temperate southern hemisphere paleolatitude with a seasonally cold winter [55,60]. Thus, the paleogeographic range of small-bodied coelophysoids extends from northern low tropical latitudes through temperate latitudes of both hemispheres. This is a broad latitudinal and environmental range for coelophysoids in the 20 kilogram size range. Given the lack of evidence of significantly different metabolic adaptations in these closely-related basal theropods, any biophysical scenario must satisfy both the hot and cold microclimates. While no single biophysical condition (i.e., combination of CTR/RMR) satisfies the disparity in climate regimes that *Coelophysis* inhabited, varying the amount and location of insulation covering its body solves this apparent paradox.

The addition of complete insulation coverage of filamentous structures to our modeled *Coelophysis* produced similar results to those of the much larger non-insulated *Plateosaurus*. Our results demonstrate that both *Plateosaurus* and a fully insulated *Coelophysis* would have been heat stressed in the hot Chinle microclimate, limiting their distribution to temperate and boreal latitudes or high elevations (or dense forested areas) at more equatorial latitudes as mentioned above.

In extant taxa the density of insulatory structures vary with ontogeny and by season [138,142,143]. Temperature acclimatization to both hot and cold climates is a well-documented phenomenon in birds and mammals. It has been shown that cold-acclimated birds can have greater feather density, higher resting metabolic rates, and reduced evaporative cooling compared to heat-acclimated birds of the same species [see 114].

Recent studies of growth and postnatal development in early dinosaurs (e.g., *C. bauri* and *C. rhodesiensis*) and dinosauromorphs (e.g., the silesaurid *Asilisaurus* (Nesbitt and others 2010)) suggest high variation in developmental sequence and body size at skeletal maturity was likely the ancestral condition [139,144]. This differs from the moderate to low intraspecific variation in growth seen in extant archosaurs [144]. Griffin and Nesbitt [144] suggest anomalously high variability in *Coelophysis* body size at skeletal maturity may be epigenetically controlled. Higher variability in metabolic rate and core temperature range (and thus physiological efficiencies such as digestion) is consistent with increased variability in size at skeletal maturity. This size variance combined with evidence of increased respiratory efficiency [145–147] and locomotion energetics [148–150] suggests increased RMR within basal dinosaurs (or their immediate ancestors) was linked to increasing aerobic scope, as opposed to enzymatic efficiency, parental care, or increased eurythermy [e.g., 151–153].

## Future research

With the need to document how to adapt Niche Mapper for modeling extinct organisms, we have restricted our present work to the adult states of *Coelophysis* and *Plateosaurus*, in environments in which they are known to occur. Ontogenetic shifts in physiology (e.g., declining metabolic rates with age/size), thermoregulatory behavior, or insulation (e.g., dermal or subdermal) should all be taken into account in future work. Clearly, hatchlings through sub adult members of a population must have survived to adulthood in our modeled climates, whether through physiological, environmental, or behavioral means. Likewise, phylogenetically disparate but spatiotemporally contemporaneous species should be incorporated in future studies to further test ecological hypotheses.

Mechanistic modeling of physiological and environmental conditions to test for viable physiological combinations in multiple environments is a relatively new tool for deep time applications. The simulations described above outline the primary components necessary for exploring paleophysiology in deep time with Niche Mapper. The relative effect of temperature, resting metabolic rate, core temperature range, size, and epidermal insulation are much greater than those of skin/fur/feather color, or muscle, respiration, and digestive efficiencies. We do not suggest these other parameters are trivial, rather they are more suited to 'fine tuning' the model, as seen in extant examples previously mentioned, or where circumstances are favorable for a specific extinct taxa [i.e., 102]. Niche Mapper is a powerful tool that can be leveraged to address a diverse array of evolutionary questions in deep time pertaining to paleoecological carrying capacity, paleobiogeographic distribution, and survivorship across major extinction boundaries.

## Conclusions

Mechanistic models in Niche Mapper use phylogenetically-constrained physiological parameters to determine habitable microclimates for a given taxon. Based on our results, prosauropods like *Plateosaurus* would have had a resting metabolic rate close to that of modern ratites, although we cannot rule out variance in this clade's core temperature range being intermediate to that predicted for extant ratites and squamates of their size. This is not unexpected given their phylogenetic position relative to known ectothermic and endothermic crown members

and is suggestive of an acquisition of elevated metabolic rates prior to the narrowing of core temperature ranges in defense of a stable core temperature. Similarly, we suggest *Coelophysis* was more likely to maintain a ratite-like resting metabolic rate than a monotreme or squamate-like RMR. A core temperature range intermediate to extant ratites and squamates is also suggested, similar to *Plateosaurus*. The presence of variable depth, density, and distribution of epidermal insulation would not only allow for a broader range of environmental tolerances, it appears to be a physiological necessity for *Coelophysis* and likely for most small ornithodirans as they increased their resting metabolic rates above ancestral levels.

Our results illustrate the interconnected nature of morphology, physiology, environmental variables and how they constrain organismal energetics, behavior, and geographic distribution. Niche Mapper is a flexible tool that can be applied to extinct organisms in deep time whose body shapes have no direct modern analog.

## Supporting information

**S1 Appendix. Niche mapper variables.**
(PDF)

**S2 Appendix. Model test using Varanus komodoensis on Komodo Island.**
(PDF)

**S3 Appendix. Parameterizing biophysical model dimensions for fossil vertebrates.**
(PDF)

**S4 Appendix. Sensitivity analyses and additional figures.**
(PDF)

**S1 Table. Target resting metabolic rates for classes of animals.**
(PDF)

## Acknowledgments

The authors would like to thank Aaron Kufner and Adam Fitch for discussion and productive feedback on earlier versions of the manuscript and Andrew Zaffos for help troubleshooting R code. Thanks also goes to the numerous colleagues who have acted as sounding boards and provided critical evaluations of this work as it progressed over the last few years; their efforts are greatly appreciated. A special thanks to two anonymous reviewers whose suggestions vastly improved an earlier draft of this manuscript, and the reviews of Dr. Ralf Kosma and an anonymous reviewer for their constructive comments.

## Author Contributions

**Conceptualization:** David M. Lovelace, Scott A. Hartman, Paul D. Mathewson, Benjamin J. Linzmeier, Warren P. Porter.

**Data curation:** David M. Lovelace.

**Formal analysis:** David M. Lovelace, Scott A. Hartman, Paul D. Mathewson, Benjamin J. Linzmeier, Warren P. Porter.

**Investigation:** David M. Lovelace, Scott A. Hartman, Benjamin J. Linzmeier, Warren P. Porter.

**Methodology:** David M. Lovelace, Scott A. Hartman, Paul D. Mathewson, Benjamin J. Linzmeier, Warren P. Porter.

**Project administration:** David M. Lovelace.

**Resources:** David M. Lovelace, Warren P. Porter.

**Software:** Paul D. Mathewson, Benjamin J. Linzmeier, Warren P. Porter.

**Supervision:** David M. Lovelace.

**Validation:** David M. Lovelace, Paul D. Mathewson, Benjamin J. Linzmeier, Warren P. Porter.

**Visualization:** David M. Lovelace.

**Writing – original draft:** David M. Lovelace, Scott A. Hartman, Paul D. Mathewson, Warren P. Porter.

**Writing – review & editing:** David M. Lovelace, Scott A. Hartman, Paul D. Mathewson, Benjamin J. Linzmeier, Warren P. Porter.

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
