## [Decision Letter · Decision Letter 0]

31 Dec 2019

PONE-D-19-26881

Modeling Dragons: Using linked mechanistic physiological and microclimate models to explore environmental, physiological, and morphological constraints on the early evolution of dinosaurs

PLOS ONE

Dear Dr. Lovelace,

Thank you for submitting your manuscript to PLOS ONE. After careful consideration, we feel that it has merit but does not fully meet PLOS ONE’s publication criteria as it currently stands. Therefore, we invite you to submit a revised version of the manuscript that addresses the points raised during the review process.

Reviewer 1 would be happy with a minor revision, whereas Reviewer 2 voted for rejection of your manuscript. In his detailed comments however, he has sympathy with your approach - yet has concerns that your results are in line with the fossil record. He also doubts the application of Niche Mapper on fossils. However he raised so many valuable and thoughtful points that I feel you could re-write the discussion to make readers aware of the uncertainties of both the method applied and the interpretation of the results.

A discussion should be self-critical. If you are willing to take the arguments of reviewer 2 and discuss them thoroughly, I invite you to re-write it.    

====

We would appreciate receiving your revised manuscript by Feb 14 2020 11:59PM. To enhance the reproducibility of your results, we recommend that if applicable you deposit your laboratory protocols in protocols.io, where a protocol can be assigned its own identifier (DOI) such that it can be cited independently in the future. For instructions see: http://journals.plos.org/plosone/s/submission-guidelines#loc-laboratory-protocols

We look forward to receiving your revised manuscript.

Kind regards,

Ulrich Joger

Academic Editor

PLOS ONE

2. Please include a caption for figure 12.

Reviewers' comments:

Reviewer's Responses to Questions

**Comments to the Author**

1. Is the manuscript technically sound, and do the data support the conclusions?

Reviewer #1: Yes

Reviewer #2: No

2. Has the statistical analysis been performed appropriately and rigorously? 

Reviewer #1: Yes

Reviewer #2: Yes

3. Have the authors made all data underlying the findings in their manuscript fully available?

Reviewer #1: Yes

Reviewer #2: Yes

4. Is the manuscript presented in an intelligible fashion and written in standard English?

Reviewer #1: Yes

Reviewer #2: Yes

5. Review Comments to the Author

Reviewer #1: Dear author,

this indead is a very interesting manuscript and I can see that a lot of time and work was necessary to write it. You find the corrections in the attached file. It would be interesting to test the methods in other dinosaur taxa as well in future publications. The body masses, especially of Plateosaurus, should not be estimated too high, because this seems to be quite unrealistic.

Reviewer #2: This paper uses the biophysical modelling software Niche Mapper to investigate the likely physiology and climatic tolerances of Late Triassic dinosaurs. These results are used to evaluate the hypothesis that large-bodied sauropodomorphs were precluded from lower-latitudes during the Late Triassic by high temperatures, finding that a metabolic profile permitting survivability at higher latitudes is inconsistent with the environment of the Chinle Formation. By contrast, the broad distribution of coelophysoids during the Late Triassic and Early Jurassic is used to suggest the variable development of feathery integument in these taxa, allowing them to survive a range of temperatures.

I do like what this paper is trying to achieve – establishing a quantitative, analytical protocol to test metabolic hypotheses in extinct taxa. If successful at this, Nice Mapper would undoubtedly be a very welcome addition to the palaeontologist’s toolbox. Unfortunately, however, I have multiple concerns rising from conflicts between the results of this paper and the fossil record than mean I am presently unconvinced as to whether or not the models have accurately reconstructed the physiology of these dinosaurs. In particular, from the current result I do not think that Niche Mapper would resolve any non-insulated small archosaur as viable in the Triassic of Europe at all, which is at direct odds with the fossil record. This is fatal to the paper, given that one of its main roles is introducing this method to palaeontologists, but at current it is unclear whether this method is too imprecise to actually be useful. Further, although I do think the author’s conclusions regarding thermal stratification of sauropodomorph communities and theropod integument are plausible, and indeed even reasonable, my concerns regarding the results means that it is presently unclear if they actually support these conclusions. In that case, the paper would not bring anything new to this discussion. Finally, I also have concerns regarding the polarity of discussion resulting in circular reasoning in the paper. Consequently, I must unfortunately advise rejection of the paper in its current form.

Still, I must reiterate that I do appreciate the goal of this paper, and applaud both the vision of the authors and the effort that has been put into providing a broad suite of sensitivity analyses. I do have some suggestions for additional sensitivity analyses that would help convince me that the Niche Mapper experiments could be successful in explaining Late Triassic dinosaur distribution. These are explained below, along with my detailed comments on the paper. If these are addressed, I would be more than happy to review this paper again during a subsequent submission.

 

Detailed comments

In general

A key result of this study is that an uninsulated Coelophysis would suffer extreme cold stress in temperate latitudes, obligating insulating integument. Comparison of results with Plateosaurus and Varanus suggest these differences are primarily the result of size. From the results presented I am concerned that Nice Mapper would resolve any small, uninsulated archosaur/reptile more broadly as unviable in high-latitude Triassic environments, which would be inconsistent with the fossil record. For instance, crurotarsans of broadly similar ecology and body shape (e.g. Ornithosuchus) or subequal size (and in some cases much smaller – e.g. Terrestisuchus) to Coelophysis are known from temperate Triassic communities, despite their phylogenetic position making insulating integument highly unlikely. I acknowledge that these taxa were still anatomically different from Coelophysis… but they were more similar to it than Varanus, which is nonetheless resolved to exhibit similar model performance to Coelophysis, under common metabolic inputs. I appreciate that Niche Mapper is well validated from extant taxa, but we are dealing with a very different world and fauna here. This problem is compounded by ontogeny. Whereas adult Plateosaurus exhibits broad thermal tolerance due to its size, it had to spend years (~12; Sander & Klein, 2005) as a growing juvenile first. Hence, allometric shape change, rapid growth and integument nonwithstanding, it must have been a viable organism for at least part of a year at Coelophysis-grade body sizes. Would a baby Plateosaurus be a viable organism in these analyses? It has been widely suggested that more deeply-nested sauropodomorphs exhibited dramatic shifts in metabolic rate through ontogeny: either this or ontogenetic integument change could be parts of an answer here but also highlight the massive sources of error inherent in this study.

TL; DR: For me to be convinced that these analyses are representative, I would need some kind of sensitivity analysis to prove that a small reptile could work in a high-latitude Triassic fauna. This isn’t arbitrary scepticism: there are many uncertainties in the inputs for these models, and so myriad sensitivity analyses are necessary to ensure that error bars are tight enough for them to be useful.

 

There is also bit of a disconnect between the polarity of the analyses performed and that described in the abstract, introduction and discussion. The experiments here ultimately use the distribution of Plateosaurus and coelophysoids, assuming (reasonably) that it was driven by temperature, to test between hypothesised metabolic regimes and integuments of these animals. This makes sense, as it is using observed data to inform more speculative traits. However, the abstract, introduction and discussion instead imply that metabolic data was used to test the importance of thermal stress versus other effects in structuring Triassic dinosaur distribution, when alternative hypotheses are not really evaluated. For example, Whiteside et al. (2015) suggested that metabolic requirements, rather than thermal stress, drove these patterns. The results here finding that Plateosaurus could easily make its metabolic requirements do refute this scenario, but this point does not receive adequate discussion at the moment given its importance. Other ecological/biological drivers are very hard to test, but they do need to be discussed. I know this comment on how the experiments are framed may sound like splitting hairs, but it is important to avoid circularity. Further, the discussion states that thermal stress was the “primary driver” on sauropodomorph distribution in the Late Triassic, but never makes the caveat that there are other potential drivers that are not tested here, or even readily testable at all.

Other comments:

Page 4: “For instance, it has been noted that there is an absence of large (>~1000 kg) prosauropod dinosaurs in the well-studied tropical to subtropical latitudes during the Late Triassic (e.g., the Chinle Formation of southwestern U.S.), while smaller (<~100 kg) theropod dinosaurs and their closest relatives are quite common.” – This is true, and is the whole point of the paper. However, something that is not noted throughout is that small-bodied (~10kg) sauropodomorphs (Thecodontosaurus, Pantydraco) are known from higher latitude European faunas during the Late Triassic, but are also absent from the Chinle Formation. This raises two issues. First: I doubt that Niche Mapper would resolve these taxa as viable in the formations in which they are known to have occurred (see above). Second: what was precluding them from the Chinle Formation? Whereas the paper does make an internally logical case for thermal stress excluding large sauropodomorphs from the Chinle Formation, the occurrence of these smaller-bodied taxa is still relevant as it raises the possibility of alternative forces limiting the distribution of sauropodomorphs more generally (but see comments below on trackways). At the very least, it provides a cautionary note on the suggestion in the main text that thermal stress was the “primary driver” of sauropodomorph distribution in the Late Triassic.

Page 21: “Coelophysis has long been considered a predatory theropod” – I think you can be bolder than this, given the known gut contents.

Page 24 (and supplementary information): “The parameters outlined above had relatively small effects on metabolic needs of the modeled organisms” – I applaud the range of sensitivity analyses performed, and do not have an issue with each being resolved to have a negligible overall effect. However, an estimate of the cumulative error/image of total error bars from these uncertainties would be helpful (in the supplemental information?) to help the reader gauge accuracy.

Page 45: “The daily temperature profile [for insulated Coelophysis] for the month of May with a squamate-like RMR and CTR is strikingly similar to that seen for a small adult komodo dragon (e.g., Fig. 11).” – This result seems to suggest that body shape has little overall effect on temperature profile. Is this reasonable? Or does it instead source from uncertainties in reconstructing original shape in the dinosaurs?

Pages 49: “The trackmakers would have been similar in size to the Early Jurassic skeletons of Seitaad (Sertich and Loewen 2010) and Sarahsaurus (Rowe and others 2011).” – It would be worth explicitly stating here that Seitaad and Sarahsaurus are both comparatively small (~100kg). Indeed, noting a size-dependent pattern is still notable in the Early Jurassic, with north American taxa (Anchisaurus, Sarahsaurus, Seitaad) being much smaller than the largest representatives from higher latitudes (e.g. Elliot and Lufeng Formation taxa) would only bolster your argument (although the obviously uneven sampling of these communities is a problem). Looking at your current results, I wonder if a 100kg Plateosaurus would be ok in a low-latitude setting anyway, obviating anything problematic about these trackways even before elevation is considered – modelling this may hence be worthwhile. Indeed, this would actually help your argument, as it would prevent the otherwise paradoxical occurrence of small-bodied sauropodomorphs in higher latitude faunas alone from suggesting other causal mechanisms structuring ‘prosauropod’ occurrence.

Page 50: “Large size alone is not a limiting factor for the Late Triassic Chinle paleoecosystem.” – This is true. You could, however, go a bit further by noting that, even then, the largest dicynodont taxon, Lisowicia, is known from European deposits. Shuvosaurids, which exhibit many convergences with dinsoaurs, also show a similar pattern, with small-bodied taxa from the Chinle Formation (Effigia, Shuvosaurus) but larger forms known from higher latitudes in South America (Sillosuchus). Although overall numbers of data points are quite low, discussion of these and other examples suggesting a phylogenetically broad occurrence of Bergmann’s Rule in the Late Triassic would only help to bolster your arguments.

Page 52: “and similarly sized C. rhodesiensis is well known from the Elliot Formation (Zimbabwe) which was deposited in a temperate southern hemisphere paleolatitude… inhabited, varying the amount and location of insulation covering its body solves this apparent paradox.” – Alternatively, was the Elliot Formation simply more moderate in temperature than the Lowenstein Formation? I know they were of comparable latitude, but palaeoclimatic data is not discussed. Another way of looking at this, while remaining in the same explicitly comparable Late Triassic time bin, would be to see how the Coelophysis model fares when scaled-up to a Liliensternus-sized individual? The integumentary argument already presented in the paper is reasonable, but surely the whole point of this method is that it allows us to exhaustively test these scenarios?

Page 52: “It is likely that that these structures were lost as sauropodomorphs increased in size - increased mass alone can expand tolerance of cooler temperatures and stabilize internal temperature variation, but not without its own energetic costs.” – What about Thecodontosaurus? It is a small bodied (~10kg) sauropodomorph known from 35N Late Triassic sites. Presumably, the authors hold that it would retain insulating integument as it precedes phyletic size increase within Sauropodomorpha (but see Bagualosaurus) but this does not receive any discussion, and would further highlight the numbers of assumptions still necessary to support their hypothesis. See also my comments above about juvenile Plateosaurus.

6. PLOS authors have the option to publish the peer review history of their article (what does this mean?). If published, this will include your full peer review and any attached files.

Reviewer #1: Yes: Kosma, Ralf

Reviewer #2: Yes: D J Button

---

## [Author Response · Author response to Decision Letter 0]

27 Feb 2020

This information is cut and pasted from our attached 'rebuttal letter' which is within the Cover Letter document.

From the editor:

Thank you for submitting your manuscript to PLOS ONE. After careful consideration, we feel

that it has merit but does not fully meet PLOS ONE’s publication criteria as it currently stands.

Therefore, we invite you to submit a revised version of the manuscript that addresses the points

raised during the review process.

Reviewer 1 would be happy with a minor revision, whereas Reviewer 2 voted for rejection of

your manuscript. In his detailed comments however, he has sympathy with your approach - yet

has concerns that your results are in line with the fossil record. He also doubts the application of

Niche Mapper on fossils. However he raised so many valuable and thoughtful points that I feel

you could re-write the discussion to make readers aware of the uncertainties of both the method

applied and the interpretation of the results.

A discussion should be self-critical. If you are willing to take the arguments of reviewer 2 and

discuss them thoroughly, I invite you to re-write it.

1. Please ensure that your manuscript meets PLOS ONE's style requirements, including those

for file naming. The PLOS ONE style templates can be found at

http://www.journals.plos.org/plosone/s/file?id=wjVg/PLOSOne_formatting_sample_main_body.p

df and

http://www.journals.plos.org/plosone/s/file?id=ba62/PLOSOne_formatting_sample_title_authors

_affiliations.pdf

While revising your submission, please upload your figure files to the Preflight Analysis and

Conversion Engine (PACE) digital diagnostic tool, https://pacev2.apexcovantage.com/. PACE

helps ensure that figures meet PLOS requirements. To use PACE, you must first register as a

user. Registration is free. Then, login and navigate to the UPLOAD tab, where you will find

detailed instructions on how to use the tool. If you encounter any issues or have any questions

when using PACE, please email us at figures@plos.org. Please note that Supporting

Information files do not need this step.

Done .

2. Please include a caption for figure 12.

Response: Figure 12 was mislabeled as Figure 14 (there is also a correct Figure 14);

this was changed to reflect that it is indeed Figure 12 - the caption is correct with this

change.

Reviewer Comments to the Author

Please use the space provided to explain your answers to the questions above. You may also

include additional comments for the author, including concerns about dual publication, research

ethics, or publication ethics. (Please upload your review as an attachment if it exceeds 20,000

characters)

Reviewer #1: Dear author,

this indead is a very interesting manuscript and I can see that a lot of time and work was

necessary to write it. You find the corrections in the attached file. It would be interesting to test

the methods in other dinosaur taxa as well in future publications. The body masses, especially

of Plateosaurus, should not be estimated too high, because this seems to be quite unrealistic.

We intend to do several follow up papers, thanks for the support and suggestions

Reviewer #1. We realize that 50-60 kg for Coelophysis and 1600-3000kg masses for

Plateosaurus are excessive and we do not consider them viable options, rather we are

using them to convey the effect that under or overestimating masses have on our model

- this is why we only use the 850 kg Plateosaurus and 21kg Coelophysis in our

experiments (save for the mass estimate sensitivity analyses).

In the text (under Mass Estimates) we state: “ It is unlikely that the mass exceeds

30 kg for the skeletal dimensions used for this analysis of mass estimates [for

Coelophysis ] ”. For Plateosaurus we state: “ We ran experiments assuming a total mass

of 600, 850, 1150, 1600, 2000, and 3000 kg (Fig. 6; see also S4 Figure 8). The first

three states (600-1150 kg) more likely capture a realistic mass estimate range for the

skeleton and are representative of mass estimates in the literature for this specimen

[97-100]. The last three states (1600-3000 kg) were used to observe how an extreme

(nonviable) overestimate of mass would affect the model results. ”

To make this more clear we added “( nonviable )” to this last sentence.

Additional response to Reviewer #1’s comments within the marked up MS.

All copy edit comments were accepted, unless indicated in bullet points below.

The changes are highlighted in our revised MS (as tracked changes). Specific comments

to outlined problems or questions are bulleted below.

● Page 39: Reviewer #1: “why is an insulated individual [ Coelophysis ] heat

stressed under cold climate conditions?”

○ We added “ This heat stress is overcome with a slight reduction in

insulation or a broadening of CTR (see below). ” as this is addressed

later in the discussion of insulation with a high RMR/CTR combined with

full insulation.

Reviewer #2: This paper uses the biophysical modelling software Niche Mapper to investigate

the likely physiology and climatic tolerances of Late Triassic dinosaurs. These results are used

to evaluate the hypothesis that large-bodied sauropodomorphs were precluded from

lower-latitudes during the Late Triassic by high temperatures, finding that a metabolic profile

permitting survivability at higher latitudes is inconsistent with the environment of the Chinle

Formation. By contrast, the broad distribution of coelophysoids during the Late Triassic and

Early Jurassic is used to suggest the variable development of feathery integument in these taxa,

allowing them to survive a range of temperatures.

I do like what this paper is trying to achieve – establishing a quantitative, analytical protocol

to test metabolic hypotheses in extinct taxa. If successful at this, Nice Mapper would

undoubtedly be a very welcome addition to the palaeontologist’s toolbox. Unfortunately,

however, I have multiple concerns rising from conflicts between the results of this paper and the

fossil record than mean I am presently unconvinced as to whether or not the models have

accurately reconstructed the physiology of these dinosaurs. In particular, from the current result

I do not think that Niche Mapper would resolve any non-insulated small archosaur as viable in

the Triassic of Europe at all, which is at direct odds with the fossil record.

[inserted author response] The above points are raised in the ‘detailed comments’ and

will be addressed there.

This is fatal to the paper, given that one of its main roles is introducing this method to

palaeontologists, but at current it is unclear whether this method is too imprecise to actually be

useful.

[inserted author response] The above points are raised in the ‘detailed comments’ and

will be addressed there. However, we would point out that this method is not ‘imprecise’

as Reviewer #2 states. As a mechanistic model the precision is high and reproducible

with the same input parameters, but lacking any significant means for absolute testing

we cannot at this time measure the accuracy outside of bounding it by other means such

as paleobiogeographic distribution, a phylogenetic framework, osteohistology and

isotopic thermometry (which we do).

The values we use for climate have a broad range (not super precise, but the

proxies are well vetted for accurately representing climate regimes). This is, in part, why

we use three climate regimes (the state of our understanding of ancient climates in deep

time is not highly precise, but given the error associated with climate proxies it is

reasonably accurate). This paper is meant to show that a reasonably bound set of

parameters can be used to test physiological (and behavioral) traits that provide a

solution to the model, i.e., under currently known distributions and paleoclimates of the

studied organisms (to the best of our knowledge at this time). Furthermore, the model

can generate hypotheses that are predictive and falsifiable which makes it a valuable

tool for deep time studies. At this time we cannot likely determine with high accuracy the

exact physiology of extinct organisms, but we can quantitatively constrain their probable

physiology based on known size, shape, phylogenetic position, geographic distributions,

and paleoclimate. Other proxy data such as clumped isotope analyses and histology

help further constrain the model and improve accuracy (see Padian and de Ricqlès,

2020). As new data arise, these may be incorporated to improve modeled results.

Another way to look at this is our Niche Mapper models in deep time produce a

hypothesis, much like a phylogenetic analysis produces many equally most

parsimonious hypotheses. We present all of the solutions from the model, and our

interpretations of the most parsimonious solutions. The addition of new paleogeographic,

isotopic, physiological or paleoclimatic data (additional OTU’s or character/character

states in the case of the phylogenetic analyses analogy) will likely lead to different

results supporting or rejecting previous hypotheses, and new hypotheses will be

generated until hopefully at some future point a consensus is reached. That is the power

of this model, and the point of this paper - the demonstration that combined mechanistic

models of microclimate and physiology can be used in deep time to build, support, or

reject hypotheses.

● Padian K, de Ricqlès A. 2020. Inferring the physiological regimes of extinct

vertebrates: methods, limits and framework. Phil. Trans. R. Soc. B 375:

20190147. http://dx.doi.org/10.1098/rstb.2019.0147

Further, although I do think the author’s conclusions regarding thermal stratification of

sauropodomorph communities and theropod integument are plausible, and indeed even

reasonable, my concerns regarding the results means that it is presently unclear if they actually

support these conclusions. In that case, the paper would not bring anything new to this

discussion. Finally, I also have concerns regarding the polarity of discussion resulting in circular

reasoning in the paper. Consequently, I must unfortunately advise rejection of the paper in its

current form.

[inserted author response] We will address the above statement(s) in the detailed

comments below.

Still, I must reiterate that I do appreciate the goal of this paper, and applaud both the vision of

the authors and the effort that has been put into providing a broad suite of sensitivity analyses. I

do have some suggestions for additional sensitivity analyses that would help convince me that

the Niche Mapper experiments could be successful in explaining Late Triassic dinosaur

distribution. These are explained below, along with my detailed comments on the paper. If these

are addressed, I would be more than happy to review this paper again during a subsequent

submission.

[inserted author response] We will address the potential additional sensitivity analyses in

the detailed comments below. In brief - the suggested sensitivity analyses include

modeling non-dinosaurian, and ontogenetic ranges of Plateosaurus / Coelophysis - these

are not so much ‘sensitivity’ analyses, but a suite of new experiments from the ground up

(different physiologies, behaviors, etc.) and, while a great suggestion (indeed we intend

to do just that) it is currently beyond the scope of this study.

Detailed comments

In general

A key result of this study is that an uninsulated Coelophysis would suffer extreme cold stress in

temperate latitudes, obligating insulating integument. Comparison of results with Plateosaurus

and Varanus suggest these differences are primarily the result of size. From the results

presented I am concerned that Nice Mapper would resolve any small, uninsulated

archosaur/reptile more broadly as unviable in high-latitude Triassic environments, which would

be inconsistent with the fossil record.

[inserted author response] We can appreciate the concern that Reviewer #2 has

regarding the possible success or failure to model other small archosaurs (or reptiles) at

high latitudes under the same climate conditions as outlined by our modeled Triassic

environments. We agree that IF our model could not resolve smaller reptiles that are

known from the high latitude fossil record that would call into question the reliability of

the model. However, as Reviewer #2 has noted (below) they are aware that Niche

Mapper is well vetted for a very broad range of extant organisms, including small

reptiles. Moreover, Niche Mapper readily incorporates burrowing, nesting, climbing trees,

and hibernation as a behavioral means of thermoregulation, torpor (or aestivation),

basking, etc. (Porter et al., 1973; also see “Additional Modeling References” below).

Most of the aforementioned modeled reptiles incorporated several thermoregulatory

behaviors typical of small (< 10 kg) reptiles. Behavioral traits such as these allow small

reptiles to survive hot and cold conditions that would otherwise be beyond their ability to

thermoregulate. For instance, small reptiles enter burrows (which maintain a more

moderate temperature) to cool off, stay warm, or enter a state of torpor to survive longer

periods of otherwise non-viable climatic conditions. (see “Additional Modeling

References” below).

Several of these behavioral traits were not modeled for Coelophysis or

Plateosaurus because it is unlikely that either of these dinosaurs were burrowing or

hibernating (Sanders, et al., 2004; Klein and Sanders, 2007; Griffin and Nesbitt, 2016).

That is not to say that all dinosaurs did not burrow, several high latitude dinosaurs are in

fact specialized for it (Varicchio et al., 2007; Martin, 2009). It is likely that small

reptiles/other archosaurs living alongside Plateosaurus would be viable if allowed to

burrow without invoking dermal insulation such as hair or feathers. Several clades of

high latitude Triassic vertebrates are also known to be burrowers, such as the amphibian

Broomistega , and cynodont Thrinaxodon (Fernandez et al., 2013), as well as unknown

vertebrate burrowers evidenced only by their empty burrows (Hasiotis et al., 2010;

Groenewald et al., 2001). One of us (DML) has been working a Late Triassic horizon that

preserves skeletons of a small temnospondyl amphibian in aestivation burrows

(Lovelace et al., 2017). However, modeling all of these additional organisms is beyond

the scope of our paper, which is meant primarily to demonstrate the utility and

application of this form of modeling to deep time.

● We added: “ Likewise, phylogenetically disparate but spatiotemporally

contemporaneous species should also be incorporated in future studies to

further test ecological hypotheses. ” at the end of the discussion to state

this.

Additional supporting comments are made in the discussion that caveat

what we know, and what we are lacking or need to improve upon.

References:

Sander PM, Klein N, Buffetaut E, Cuny G, Suteethorn V, Loeuff JL. Adaptive radiation in

sauropod dinosaurs: bone histology indicates rapid evolution of giant body size through

acceleration. Org Divers Evol. 2004; 4(3): 165-173.

Klein N, Sander PM. Bone histology and growth of the prosauropod dinosaur Plateosaurus

engelhardti von Meyer, 1837 from the Norian bonebeds of Trossingen (Germany) and

Frick (Switzerland). Spec Pap Paleontol. 2007; 77: 169-206.

Griffin CT, Nesbitt, SJ. The histology and femoral ontogeny of the Middle Triassic (?late Anisian)

dinosauriform Asilisaurus kongwe and implications for the growth of early dinosaurs. J

Vertebr Paleontol. 2016; 36(3): e1111224.

Porter, W. P., J. W. Mitchell, W. A. Beckman and C. B. DeWitt (1973). "Behavioral

implications of mechanistic ecology." Oecologia 13(1): 1-54.

Varricchio D, Martin AJ, and Katsura Y. First trace and body fossil evidence of a burrowing,

denning dinosaur. Proc. R. Soc. B (2007) 274, 1361–1368

Martin, AJ. Dinosaur burrows in the Otway Group (Albian) of Victoria, Australia, and their

relation to Cretaceous polar environments. Cretaceous Research; 30: ) 1223–1237

Fernandez V, Abdala F, Carlson KJ, Cook DC, Rubidge BS, et al. (2013) Synchrotron

Reveals Early Triassic Odd Couple: Injured Amphibian and Aestivating Therapsid Share

Burrow. PLoS ONE 8(6): e64978. doi:10.1371/journal.pone.0064978

Stephen T. Hasiotis , Robert W. Wellner , Anthony J. Martin & Timothy M. Demko (2004)

Vertebrate Burrows from Triassic and Jurassic Continental Deposits of North America

and Antarctica: Their Paleoenvironmental and Paleoecological Significance, Ichnos,

11:1-2, 103-124, DOI: 10.1080/10420940490428760

Groenewald GH, Welman J, MacEachern JA (2001) Vertebrate burrow complexes from

the Early Triassic Cynognathus Zone (Driekoppen Formation, Beaufort Group) of the

Karoo Basin, South Africa. Palaios 16: 148–160.

D. Lovelace, A. K. Huttenlocker, J. D. Pardo, A. M. Kufner, G. Chen, K. Li. The first Late

Triassic temnospondyl mass-mortality localities from the Popo Agie Formation, Fremont

County, WY. 2017. Soc. Vert Paleo, Abstract pg. 153.

For instance, crurotarsans of broadly similar ecology and body shape (e.g. Ornithosuchus) or

subequal size (and in some cases much smaller – e.g. Terrestisuchus) to Coelophysis are

known from temperate Triassic communities, despite their phylogenetic position making

insulating integument highly unlikely. I acknowledge that these taxa were still anatomically

different from Coelophysis… but they were more similar to it than Varanus, which is nonetheless

resolved to exhibit similar model performance to Coelophysis, under common metabolic inputs.

Agreed - Ornithosuchus and Terrestisuchus are more similar in ecology/body

shape than Varanus but there is actualistic data for Varnus which is why we included it

as a cursory test for the modeling of a large-bodied squamate (to ensure Coelophysis

and Plateosaurus modeled as squamates would be reasonably accurate).

Again, the suggestion to model crocodylomorphs is a great idea and well worth

pursuing in a followup paper(s). It would be quite telling to quantify the benefit a shallow,

moderate, and deep burrow might provide a rather gracile Terrestrisuchus compared to

an individual without the benefit of a burrow. In fact, it would be a great future project to

model Alligator geographic distribution with and without the ability to burrow/den. It is

certainly well established that Alligator (and Varanus ) use dening/burrowing to survive

wildfires, and extreme weather such as significant cold spells and hurricanes (I would

speculate at this time that without the ability to den it would not likely survive nearly as

far north as it currently does), as well as aqueous diving behaviors to thermoregulate in

extreme heat; these fossorial habits may well be a holdover from their crocodylomorph

ancestors, e.g:

“The occurrence of almost complete articulated skeletons in nearby correlated

stratigraphic levels suggests they could dig large, deep burrows in soft substrates

that allowed, like living alligators, thermoregulation. These excavated holes

provide a source of water during dry periods. This ethological aspect probably

allowed them to live in more terrestrial and arid environments.” - I.S. Carvalho et

al., 2005, Baurusuchus salgadoensis , a New crocodylomorpha from the Bauru

Basin (Cretaceous), Brazil. Gondwana Research; 8(1): 11-30.

It should be noted too, that while Terrestrisuchus and its relatives may not have

had epidermal insulation (currently phylogenetically constrained to Ornithodira), either

croc-line archosaurs or archosaurs in general may have had elevated resting metabolic

rates based on respiratory anatomy (Brocklehurst and others, 2020), upright locomotion

(Seymour and others, 2004) and inferred growth rates (Legendre and others, 2016).

Investigating the impact of those metabolic inferences on non-dinosaurian archosaurs is

an exciting avenue of potential research, for ourselves and others who adopt

mechanistic modeling techniques.

References:

Brocklehurst, Robert J., Emma R. Schachner, Jonathan R. Codd, and William I. Sellers.

"Respiratory evolution in archosaurs." Philosophical Transactions of the Royal Society B

375, no. 1793 (2020): 20190140.

Legendre, Lucas J., Guillaume Guénard, Jennifer Botha-Brink, and Jorge Cubo.

"Palaeohistological evidence for ancestral high metabolic rate in archosaurs." Systematic

Biology 65, no. 6 (2016): 989-996.

Seymour, Roger S., Christina L. Bennett-Stamper, Sonya D. Johnston, David R. Carrier,

and Gordon C. Grigg. "Evidence for endothermic ancestors of crocodiles at the stem of

archosaur evolution." Physiological and Biochemical Zoology 77, no. 6 (2004):

1051-1067.

I appreciate that Niche Mapper is well validated from extant taxa, but we are dealing with a very

different world and fauna here.

Niche Mapper does not assume any particular physiology, anatomy, or

environment, and its inputs can be made as different or similar to modern examples as

the evidence allows. Beyond that it is simply calculating thermodynamic processes,

which surely were not significantly different in the Triassic than they are today. On the

contrary, Niche Mapper provides a tool for quantitatively testing how different taxa and

environments could have been. For instance, RMR and CTR combinations outside of

those observed in modern organisms can be quantitatively tested against

paleobiogeography and paleoenvironmental datasets. Conversely, paleobiogeography

and paleophysiology can be tested against paleoclimate data to derive quantitative

estimates, and the impacts of parameters that we currently have no proxies for (i.e.

cloud cover). While the climates, flora and fauna of the Triassic were different from

today, there is little reason to think those differences extended to cellular function, or

that local climates could have had conditions physically impossible in the modern world.

This problem is compounded by ontogeny. Whereas adult Plateosaurus exhibits broad thermal

tolerance due to its size, it had to spend years (~12; Sander & Klein, 2005) as a growing

juvenile first. Hence, allometric shape change, rapid growth and integument notwithstanding, it

must have been a viable organism for at least part of a year at Coelophysis-grade body sizes.

Would a baby Plateosaurus be a viable organism in these analyses? It has been widely

suggested that more deeply-nested sauropodomorphs exhibited dramatic shifts in metabolic

rate through ontogeny: either this or ontogenetic integument change could be parts of an

answer here but also highlight the massive sources of error inherent in this study.

[inserted author response] This point has been taken into consideration and we added

“ Ontogenetic shifts in physiology (e.g., declining metabolic rates with age/size),

thermoregulatory behavior, or insulation (e.g., dermal or subdermal) should all be

taken into account in future work. Clearly, hatchlings through sub adult members

of a population must have survived to adulthood in our modeled climates,

whether through physiological, environmental, or behavioral means. Likewise,

phylogenetically disparate but spatiotemporally contemporaneous species should

also be incorporated in future studies to further test ecological hypotheses. ” to

the end of the discussion.

Ontogenetic shifts in physiology are certainly of interest to us, and in fact is a

project underway (among others that relate to additional, non-dinosaurian fauna

of the Late Triassic). However, whether we model a juvenile or not does not

change our current results nor their interpretation. Modeling a juvenile of either of

these species will provide a great source of information for that specific model -

we can then test for variability in insulation, metabolic rate, behavior, etc.

throughout ontogeny. This is critical to our understanding of paleobiology - but

not mandatory to resolve this issue in this paper. The fact also remains, the

model is well suited to test these problems, and that is the point of this paper -

we can model extinct organisms and quantify the differences allometry, variable

met rate, etc. have throughout ontogeny. Our model supports an adult

Plateosaurus at high latitudes, but it is excluded at low latitudes due to heat

stress. All observational data are satisfied for an adult.

It is known that juvenile (nestling) sauropodomorphs such as Mussaurus were

not only allometrically different than adults they were functionally different as

obligate quadrupeds that only later in ontogeny gained a bipedal stance (Otero et

al., 2019). It is likely that there are significant differences that would need to be

accounted for when modeling a juvenile vs. adult form.

● Otero, Alejandro; Cuff, Andrew R.; Allen, Vivian; Sumner-Rooney, Lauren;

Pol, Diego; Hutchinson, John R. (2019-05-20). "Ontogenetic changes in

the body plan of the sauropodomorph dinosaur Mussaurus patagonicus

reveal shifts of locomotor stance during growth". Scientific Reports. 9 (1):

7614.

TL; DR: For me to be convinced that these analyses are representative, I would need some kind

of sensitivity analysis to prove that a small reptile could work in a high-latitude Triassic fauna.

This isn’t arbitrary scepticism: there are many uncertainties in the inputs for these models, and

so myriad sensitivity analyses are necessary to ensure that error bars are tight enough for them

to be useful.

This is addressed above - there is a broad body of research (see Additional

Modeling References below) that support small reptiles in a variety of climates with high

accuracy. The problem isn’t that a small reptile (or juvenile dinosaur) wouldn’t work in a

high latitude model - they do - it’s just that they are inherently different with different

solutions to thermoregulation than an adult Coelophysis or Plateosaurus .

There is also bit of a disconnect between the polarity of the analyses performed and that

described in the abstract, introduction and discussion. The experiments here ultimately use the

distribution of Plateosaurus and coelophysoids, assuming (reasonably) that it was driven by

temperature, to test between hypothesised metabolic regimes and integuments of these

animals. This makes sense, as it is using observed data to inform more speculative traits.

However, the abstract, introduction and discussion instead imply that metabolic data was used

to test the importance of thermal stress versus other effects in structuring Triassic dinosaur

distribution, when alternative hypotheses are not really evaluated.

We appreciate this critique and have restructured the Abstract (significantly) and

Introduction (minorly) to reflect this. The addition of “ The paleobiogeographic

distribution of Plateosaurus and Coelophysis along with associated paleoclimate

proxies from known fossil localities provide boundary conditions to test hypotheses of

environmental exclusion due to physiological limitations. ” in the last paragraph aligns

the polarity of the introduction for a clearer flow.

The discussion is reworked as well, incorporating several of the suggestions and

critiques outlined throughout the reviewers comments.

Please see the ‘marked up’ version of the MS for changes - mostly they were

reformatting the order of statements to make a more consistent and logical flow as

suggested.

For example, Whiteside et al. (2015) suggested that metabolic requirements, rather than

thermal stress, drove these patterns. The results here finding that Plateosaurus could easily

make its metabolic requirements do refute this scenario, but this point does not receive

adequate discussion at the moment given its importance. Other ecological/biological drivers are

very hard to test, but they do need to be discussed. I know this comment on how the

experiments are framed may sound like splitting hairs, but it is important to avoid circularity.

Further, the discussion states that thermal stress was the “primary driver” on sauropodomorph

distribution in the Late Triassic, but never makes the caveat that there are other potential drivers

that are not tested here, or even readily testable at all.

We added: “ Whiteside and others [28] argued for climate driven

environmental and vegetative instability as a dominant factor for the exclusion of

large dinosaurs from the Late Triassic tropics. Our results suggest heat stress

alone should have been a significant barrier for Plateosaurus-sized dinosaurs.

We suggest environmental temperatures limited large bodied prosauropods from

greater appearance in tropical latitudes during the Late Triassic.”

Other comments:

Page 4: “For instance, it has been noted that there is an absence of large (>~1000 kg)

prosauropod dinosaurs in the well-studied tropical to subtropical latitudes during the Late

Triassic (e.g., the Chinle Formation of southwestern U.S.), while smaller (<~100 kg) theropod

dinosaurs and their closest relatives are quite common.” – This is true, and is the whole point of

the paper.

However, something that is not noted throughout is that small-bodied (~10kg)

sauropodomorphs (Thecodontosaurus, Pantydraco) are known from higher latitude European

faunas during the Late Triassic, but are also absent from the Chinle Formation. This raises two

issues. First: I doubt that Niche Mapper would resolve these taxa as viable in the formations in

which they are known to have occurred (see above).

We address this point above (e.g., smaller dinosaurs/reptiles).

Second: what was precluding them from the Chinle Formation? Whereas the paper does

make an internally logical case for thermal stress excluding large sauropodomorphs from the

Chinle Formation, the occurrence of these smaller-bodied taxa is still relevant as it raises the

possibility of alternative forces limiting the distribution of sauropodomorphs more generally (but

see comments below on trackways). At the very least, it provides a cautionary note on the

suggestion in the main text that thermal stress was the “primary driver” of sauropodomorph

distribution in the Late Triassic.

We added “ It should also be noted that there are no small bodied prosauropod

species known from the tropics, in contrast to several taxa known from

temperate latitudes (i.e., Thecodontosaurus (Morris, 1843) and(Galton and

others, 2007)). Our preliminary results from isometrically scaled ‘juvenile’

Plateosaurus demonstrate 10 to 100 kg individuals are viable in the hot

microclimate under the same physiological parameters (among others) as the

viable adult form modeled in the cold microclimate. This is consistent with the

presence of Late Triassic prosauropod-like tracks and ~100 kg Early Jurassic

body fossils. Although we maintain that heat stress is a limiting factor to large

bodied prosauropods in the tropics it is possible other factors such as

insurmountable paleogeographic or environmental barriers or preservational

biases account for the paucity of smaller morphs within the Chinle depocenter.”

to the discussion on prosauropod exclusion at tropical latitudes to highlight this

yet unexplained absence.

Page 21: “Coelophysis has long been considered a predatory theropod” – I think you can be

bolder than this, given the known gut contents.

We changed a citation [86] to make a stronger case and reference the 2006 paper that

demonstrates Coelophysis consumed a crocodylomorph ( Hesperosuchus ).

Page 24 (and supplementary information): “The parameters outlined above had relatively small

effects on metabolic needs of the modeled organisms” – I applaud the range of sensitivity

analyses performed, and do not have an issue with each being resolved to have a negligible

overall effect. However, an estimate of the cumulative error/image of total error bars from these

uncertainties would be helpful (in the supplemental information?) to help the reader gauge

accuracy.

As mentioned above, Niche Mapper itself is highly precise - potential sources of error

stem from inputs, not from calculations. While some sources of input error are potentially

linked (e.g. insolation and precipitation) others are not (e.g. precipitation and mass

estimates). As a result there isn’t a simple additive form of error; e.g. if we find an

estimated viable core body temperature range of 32-38 Celsius, there’s no precise way

to quantify a +/- 2 degree “error bar”. Instead, the goal of our sensitivity analyses is to

narrow the range of inputs likely to be most productively tested by future researchers.

The presence or absence of epidermal insulation is one such input, as it has a major

impact on model results (and thus we included a range of insulatory coverings in our

analyses). The coloration of any potential epidermal insulation, however, has a very

small impact on model results, and is therefore a less important avenue of research

when gauging input data. The following ‘Lung Efficiency’ figure is an example of a

sensitivity analysis that tests a range of inputs for O2 extraction efficiency - there are no

errors for each value of that range - just a calculated answer. We test the range to see

the overall effect that values has - in this case very little over all effect.

Page 45: “The daily temperature profile [for insulated Coelophysis] for the month of May with a

squamate-like RMR and CTR is strikingly similar to that seen for a small adult komodo dragon

(e.g., Fig. 11).” – This result seems to suggest that body shape has little overall effect on

temperature profile. Is this reasonable? Or does it instead source from uncertainties in

reconstructing original shape in the dinosaurs?

At the level of precision necessary for Niche Mapper analysis there are few

uncertainties in reconstructing the original shape of the dinosaurs (see supplemental

appendix 3). We do not think this similarity suggests body shape (or stance) is of little

importance; while there is similarity in the gross shape of the curve, the actual rates of

heat increase are different, with the more gracile Coelophysis heating at a quicker rate.

Pages 49: “The trackmakers would have been similar in size to the Early Jurassic skeletons of

Seitaad (Sertich and Loewen 2010) and Sarahsaurus (Rowe and others 2011).” – It would be

worth explicitly stating here that Seitaad and Sarahsaurus are both comparatively small

(~100kg). Indeed, noting a size-dependent pattern is still notable in the Early Jurassic, with

north American taxa (Anchisaurus, Sarahsaurus, Seitaad) being much smaller than the largest

representatives from higher latitudes (e.g. Elliot and Lufeng Formation taxa) would only bolster

your argument (although the obviously uneven sampling of these communities is a problem).

Looking at your current results, I wonder if a 100kg Plateosaurus would be ok in a low-latitude

setting anyway, obviating anything problematic about these trackways even before elevation is

considered – modelling this may hence be worthwhile. Indeed, this would actually help your

argument, as it would prevent the otherwise paradoxical occurrence of small-bodied

sauropodomorphs in higher latitude faunas alone from suggesting other causal mechanisms

structuring ‘prosauropod’ occurrence.

We did add “ Our preliminary results from isometrically scaled ‘juvenile’

Plateosaurus demonstrate 10 to 100 kg individuals are viable in the hot

microclimate under the same physiological parameters (among others) as the

viable adult form modeled in the cold microclimate. This is consistent with the

presence of Late Triassic prosauropod-like tracks and ~100 kg Early Jurassic

body fossils.”

Page 50: “Large size alone is not a limiting factor for the Late Triassic Chinle paleoecosystem.”

– This is true. You could, however, go a bit further by noting that, even then, the largest

dicynodont taxon, Lisowicia, is known from European deposits. Shuvosaurids, which exhibit

many convergences with dinosaurs, also show a similar pattern, with small-bodied taxa from the

Chinle Formation (Effigia, Shuvosaurus) but larger forms known from higher latitudes in South

America (Sillosuchus). Although overall numbers of data points are quite low, discussion of

these and other examples suggesting a phylogenetically broad occurrence of Bergmann’s Rule

in the Late Triassic would only help to bolster your arguments.

This is a great point, and something we hope to address in future WRT Bergmann’s

Rule and the fossil record. We feel that including a broader discussion on this topic

would take up more space than warranted for this paper.

Page 52: “and similarly sized C. rhodesiensis is well known from the Elliot Formation

(Zimbabwe) which was deposited in a temperate southern hemisphere paleolatitude…

inhabited, varying the amount and location of insulation covering its body solves this apparent

paradox.” – Alternatively, was the Elliot Formation simply more moderate in temperature than

the Lowenstein Formation? I know they were of comparable latitude, but palaeoclimatic data is

not discussed. Another way of looking at this, while remaining in the same explicitly comparable

Late Triassic time bin, would be to see how the Coelophysis model fares when scaled-up to a

Liliensternus-sized individual? The integumentary argument already presented in the paper is

reasonable, but surely the whole point of this method is that it allows us to exhaustively test

these scenarios?

We added citations that support a seasonly cold winter in the Elliot Formation

( Skeletal remains of C. bauri (our model) are well known from Chinle deposits

best represented by our hot microclimates, and similarly sized C. rhodesiensis is

well known from the Elliot Formation (Zimbabwe) which was deposited in a

temperate southern hemisphere paleolatitude with a seasonally cold winter

[55,60]. )

We also ran preliminary simulations of a 100 kg Coelophysis (cf. Liliensternus )

and the results are similar to the 21 kg Coelophysis in the moderate climate - in effect

the increase in size produces similar results to a smaller animal shifted to a slightly

warmer climate e.g. a 100 kg Coelophysis in the cold microclimate is very similar to a

21 kg Coelophysis in the moderate climate (all physiological options remaining the

same). So yes, size has a benefit as would be expected with cooler (higher latitude)

climates under Bergmann’s Rule.

In addition we ran preliminary simulations of a 10, 20, 100, 400 kg

Plateosaurus (isometrically scaled to an appropriate size to maintain the assigned

density for each body part) -as mentioned previously in this rebuttal. There are several

references to these preliminary studies in the Discussion section:

Page 52: “It is likely that that these structures were lost as sauropodomorphs increased in size -

increased mass alone can expand tolerance of cooler temperatures and stabilize internal

temperature variation, but not without its own energetic costs.” – What about

Thecodontosaurus? It is a small bodied (~10kg) sauropodomorph known from 35N Late Triassic

sites. Presumably, the authors hold that it would retain insulating integument as it precedes

phyletic size increase within Sauropodomorpha (but see Bagualosaurus) but this does not

receive any discussion, and would further highlight the numbers of assumptions still necessary

to support their hypothesis. See also my comments above about juvenile Plateosaurus.

These comments are addressed several times above where we discuss

preliminary results of isometric ‘juvenile’ Plateosaurus as well as the absence of small

~10kg prosauropods at tropical latitudes.

“It should also be noted that there are no small bodied prosauropod species

known from the tropics, in contrast to several taxa known from temperate latitudes (i.e.,

Thecodontosaurus (Morris 1843) and Pantydraco (Galton and others 2007)). Our

preliminary results from isometrically scaled ‘juvenile’ Plateosaurus demonstrate 10 to

100 kg individuals are viable in the hot microclimate under the same physiological

parameters (among others) as the viable adult form modeled in the cold microclimate.

This is consistent with the presence of Late Triassic prosauropod-like tracks and ~100

kg Early Jurassic body fossils. Although we maintain that heat stress is a limiting factor

to large bodied prosauropods in the tropics it is possible other factors such as

insurmountable paleogeographic or environmental barriers or preservational biases

account for the paucity of smaller morphs within the Chinle depocenter. “

ADDITIONAL MODELING REFERENCES:

Adolph, S. C. and W. P. Porter (1993). "Temperature, Activity, and Lizard Life Histories." The

American Naturalist 142(2): 273-295.

Adolph, S. C. and W. P. Porter (1996). "Growth, seasonality, and lizard life histories: age and

size at maturity." Oikos: 267-278.

Christian, K., C. R. Tracy and W. P. Porter (1983). "Seasonal shifts in body temperature and

use of microhabitats by Galapagos land iguanas (Conolophus pallidus)." Ecology 64(3):

463-468.

Dunham, A. E., K. L. Overall, W. P. Porter and C. A. Forster (1989). "Implications of ecological

energetics and biophysical and developmental constraints for life-history variation in dinosaurs."

Geological Society of America Special Papers 238: 1-20.

Grant, B. W. and W. P. Porter (1992). "Modeling global macroclimatic constraints on ectotherm

energy budgets." American Zoologist 32(2): 154-178.

Harlow, H. J., D. Purwandana, T. S. Jessop and J. A. Phillips (2010). "Body temperature and

thermoregulation of Komodo dragons in the field." Journal of Thermal Biology 35(7): 338-347.

Huang, S.-P., K.-W. Hung, H.-C. Fan, T.-E. Lin and R. Richard (2019). "Temperature rise

curtails activity period predicted for a winter-active forest lizard, Scincella formosensis, from

subtropical areas in Taiwan." Journal of Thermal Biology: 102475.

Huang, S.-P., W. P. Porter, M.-C. Tu and C.-R. Chiou (2014). "Forest cover reduces thermally

suitable habitats and affects responses to a warmer climate predicted in a high-elevation lizard."

Oecologia 175(1): 25-35.

Huang, S. P., C. R. Chiou, T. E. Lin, M. C. Tu, C. C. Lin and W. P. Porter (2013). "Future

advantages in energetics, activity time, and habitats predicted in a high-altitude pit viper with

climate warming." Functional Ecology 27(2): 446-458.

Jones, S. M., R. E. Ballinger and W. P. Porter (1987). "Physiological and Environmental

Sources of Variation in Reproduction: Prairie Lizards in a Food Rich Environment." Oikos 48(3):

325-335.

Kearney, M., R. Shine and W. P. Porter (2009). "The potential for behavioral thermoregulation to

buffer “cold-blooded” animals against climate warming." Proceedings of the National Academy

of Sciences 106(10): 3835-3840.

Mathewson, P. D., L. Moyer-Horner, E. A. Beever, N. J. Briscoe, M. Kearney, J. M. Yahn and

W. P. Porter (2016). "Mechanistic variables can enhance predictive models of endotherm

distributions: the American pika under current, past, and future climates." Global Change

Biology 23(3): 1048-1064.

Mitchell, N., M. R. Hipsey, S. Arnall, G. McGrath, H. B. Tareque, G. Kuchling, R. Vogwill, M.

Sivapalan, W. P. Porter and M. R. Kearney (2012). "Linking eco-energetics and eco-hydrology

to select sites for the assisted colonization of Australia’s rarest reptile." Biology 2(1): 1-25.

Mitchell, N. J., M. R. Kearney, N. J. Nelson and W. P. Porter (2008). "Predicting the fate of a

living fossil: how will global warming affect sex determination and hatching phenology in

tuatara?" Proceedings of the Royal Society of London B: Biological Sciences 275(1648):

2185-2193.

Moyer-Horner, L., P. D. Mathewson, G. M. Jones, M. R. Kearney and W. P. Porter (2015).

"Modeling behavioral thermoregulation in a climate change sentinel." Ecology and Evolution

5(24): 5810-5822.

Norris, K. S. (1967). Color adaptation in desert reptiles and its thermal relationships. Lizard

ecology: a symposium, University of Missouri Press Columbus, Missouri.

Porter, W. P. (1989). "New animal models and experiments for calculating growth potential at

different elevations." Physiological Zoology 62(2): 286-313.

Porter, W. P., J. W. Mitchell, W. A. Beckman and C. B. DeWitt (1973). "Behavioral implications

of mechanistic ecology." Oecologia 13(1): 1-54.

Porter, W. P., J. L. Sabo, C. R. Tracy, O. Reichman and N. Ramankutty (2002). "Physiology on

a landscape scale: plant-animal interactions." Integrative and Comparative Biology 42(3):

431-453.

Smith, K. R., V. Cadena, J. A. Endler, M. R. Kearney, W. P. Porter and D. Stuart-Fox (2016).

"Color Change for Thermoregulation versus Camouflage in Free-Ranging Lizards." The

American Naturalist 188(6): 668-678.

Wang, Y., W. Porter, P. D. Mathewson, P. A. Miller, R. W. Graham and J. W. Williams (2018).

"Mechanistic modeling of environmental drivers of woolly mammoth carrying capacity declines

on St. Paul Island." Ecology 99(12): 2721-2730.

---

## [Decision Letter · Decision Letter 1]

31 Mar 2020

PONE-D-19-26881R1

Modeling Dragons: Using linked mechanistic physiological and microclimate models to explore environmental, physiological, and morphological constraints on the early evolution of dinosaurs

PLOS ONE

Dear Dr. Lovelace,

Thank you for submitting your manuscript to PLOS ONE. After careful consideration, we feel that it has merit but does not fully meet PLOS ONE’s publication criteria as it currently stands. Therefore, we invite you to submit a revised version of the manuscript that addresses the points raised during the review process.

Both reviewers have only minor coments left to finalize the manuscript. THe main task for you is to include in the discussion the points raised by reviewer 2 (see his comments below).

We would appreciate receiving your revised manuscript by May 15 2020 11:59PM. To enhance the reproducibility of your results, we recommend that if applicable you deposit your laboratory protocols in protocols.io, where a protocol can be assigned its own identifier (DOI) such that it can be cited independently in the future. For instructions see: http://journals.plos.org/plosone/s/submission-guidelines#loc-laboratory-protocols

We look forward to receiving your revised manuscript.

Kind regards,

Ulrich Joger

Academic Editor

PLOS ONE

Reviewers' comments:

Reviewer's Responses to Questions

**Comments to the Author**

1. If the authors have adequately addressed your comments raised in a previous round of review and you feel that this manuscript is now acceptable for publication, you may indicate that here to bypass the “Comments to the Author” section, enter your conflict of interest statement in the “Confidential to Editor” section, and submit your "Accept" recommendation.

Reviewer #1: (No Response)

Reviewer #2: (No Response)

2. Is the manuscript technically sound, and do the data support the conclusions?

Reviewer #1: Yes

Reviewer #2: Partly

3. Has the statistical analysis been performed appropriately and rigorously? 

Reviewer #1: Yes

Reviewer #2: Yes

4. Have the authors made all data underlying the findings in their manuscript fully available?

Reviewer #1: Yes

Reviewer #2: Yes

5. Is the manuscript presented in an intelligible fashion and written in standard English?

Reviewer #1: Yes

Reviewer #2: Yes

6. Review Comments to the Author

Reviewer #1: Dear authors, there are only a very few errors found in this second review of your manuscript. Altogether the contents of the articel are very interesting, though they are very theoretically. But as you wrote from the very beginning it is a manuscript dealing with an attempt of "modeling". Therefore I interprete it as a kind of thought stimulation. The reader has to decide how far the described scenarios might be realistic or not. To me the manuscript seems to be well elaborated and the results draw a picture that is absolutely comprehensible.

Reviewer #2: In my previous review of this manuscript, I essentially only had one problem with it: I was concerned that the results within were inconsistent with allowing small-bodied archosaurs to have existed at high latitudes during the Triassic, which is at clear odds with the fossil record. I hence requested an additional analysis to prove to me that this was not the case, which would have allowed me to throw my full support behind the results.

In this resubmission, the authors have not included this analysis, as they argue that modelling additional taxa is beyond the scope of this study. Although I still think this would have been an effective way to increase confidence in the results and conclusions, I actually do take their point on this. Further, I reiterate that I appreciate that NicheMapper is well validated on extant taxa (but would add that the authors should not be surprised by additional scepticism when a method is extrapolated far beyond its test dataset). I also think that the rewriting of the paper has helped considerably to address some of my previous concerns regarding the polarity of the discussion.

Consequently, I am satisfied to recommend this paper for publication without this additional analysis. In its place, though, I really do think that the Discussion needs to contain an explicit defence of how animals broadly similar to Coelophysis and/or small prosauropods would have survived at higher latitudes. I doubt that I would be the only reader to draw these questions, and so it would only strengthen the paper to tackle them head-on. The good news is that the authors have already done so forcefully in their responses to me, citing a range of behavioural strategies, with some fossil evidence, as well as NicheMapper studies on extant taxa – incorporating this into the Discussion would only help the manuscript. I think that explicitly (if briefly) detailing the alternatives and corollaries here – that these compensatory mechanisms were widespread and/or integument may have been broadly distributed in Archosauria (albeit with the caveat that further testing is required in both of these instances) would really help give the reader what they need to evaluate the conclusions.

In summary, although I am disappointed to see some of the analyses I requested were not performed, I am prepared to admit that they are beyond the immediate scope of this contribution. Hence, I am satisfied with the paper pending expansion of the Discussion, including content from some of your responses to me, as outlined above. Were this to be completed, I would not feel that I would need to see this manuscript again, although would be happy to do so if the Editor deemed it necessary.

Specific responses to some of the author's replies to my previous comments are given in the attached reviewer's responses document.

7. PLOS authors have the option to publish the peer review history of their article (what does this mean?). If published, this will include your full peer review and any attached files.

Reviewer #1: No

Reviewer #2: No

---

## [Author Response · Author response to Decision Letter 1]

14 Apr 2020

Reviewer 1: (typographical errors in attached document)

We corrected all typographical errors outlined by Reviewer 1.

Reviewer 2 had these primary comments: They wished to see a statement that smaller taxa existed alongside the modeled organisms and that how their behavior might affect the model results.

“by stating that we know smaller-bodied reptiles existed in these

environments, but would have survived via these behaviours, as seen in extant taxa (citing a NicheMapper study of a high-latitude reptile would be useful here).”

“Again, though, it would be worth citing the broad distribution of burrowing behaviour in Triassic taxa in the discussion to help bolster your case.”

“I also think that these suggestions for how these taxa could have coped with such conditions – behavioural or integumentary – are plausible, and so should be mentioned. The content in your reply here is valuable, so why not use it in the discussion itself? I do not think I would be the only reader who would be left wondering about the repercussions for other small-bodied taxa and, even if modelling them is beyond the scope of this study, you could at least briefly address this by commenting that even though similar taxa did occur in those localities, they may have used these different strategies.”

Our additions to the MS Discussion address reviewers 1 request and are as follows:

To the ‘Deep-time model uncertainty’ section we added:

Moreover, our modeled organisms are adult forms of specific clades and are not representative of the ecosystem as a whole. Nor do we assume that juveniles from hatchling to subadult will be modeled with similar parameters as each other, or their adult counterpart - each ontogenetic stage may have different combinations of physiology, shape, behavior, insulation and niche availability. It is beyond the scope of this study to model each stage of ontogeny. To further explore and test the viability of our models future work will need to generate ontogenetic series for each species, and contemporaneous taxa at high latitudes such as small archosaurs, amphibians, and lepidosaurs known to co-occur with Plateosaurus and Coelophysis.

To the ‘Plateosaurus’ section we added:

Modeling allometrically smaller individuals coupled to growth rates [111,112] is a necessary next step. It is known that juvenile (nestling) sauropodomorphs such as Mussaurus were not only allometrically different than adults (e.g., proportionally taller skulls with short snout and larger eyes, and shorter tails and necks), they were functionally different as obligate quadrupeds that only later in ontogeny gained a bipedal stance [115, 116]. It is likely that there are significant differences that will need to be accounted for when modeling juvenile vs. adult forms.

To the ‘Integrating model results with the fossil record’ section we added:

Other behaviors such as burrowing or hibernation were excluded in our study as they are unlikely means of temperature regulation for adult Coelophysis and Plateosaurus. However, several smaller dinosaurs from temperate latitudes later in the Mesozoic are known or suspected burrowers [127,128]. It is not outside the realm of possibility that juvenile Plateosaurus (or other small prosauropods such as the 15 kg Thecodontosaurus) would have leveraged the benefits of burrowing. Fossorial behavior would allow for exploitation of more stable microhabitats in environments with higher variance in daily or annual air temperature. It is well known that small squamates, as well as the large Varanus and crocodylian Alligator use burrowing behaviors to thermoregulate, surviving extreme weather events and even wildfires [40,129,130].

A potential discrepancy with our results is the presence of small-medium bodied taxa contemporaneous with Plateosaurus such as Thecodontosaurus, early turtles (i.e, Proganochelys), and gracile crocodylomorphs (i.e., Terrestrisuchus) which would likely model as cold stressed during the winter nights if provided similar behavior parameters without the benefit of dermal insulation or greater size. These high latitude species almost certainly employed alternate thermoregulatory strategies such as burrowing, aestivation, or hibernation, much like their modern descendants. It may be that denning (burrowing) behavior was an ancestral state for crocodyliformes [131]. It should be noted too, that while Terrestrisuchus and its relatives may not have had epidermal insulation (currently phylogenetically constrained to Ornithodira), either croc-line archosaurs or archosaurs in general may have had elevated resting metabolic rates based on respiratory anatomy [132], upright locomotion [133] and inferred growth rates [134]. Investigating the impact of those metabolic and behavioral inferences on non-dinosaurian archosaurs is an exciting avenue of potential research, for ourselves and others who adopt mechanistic modeling techniques.

To the ‘Insulation in Triassic theropods’ section we added a 2020 reference we just became aware of regarding the early evolution of and added it to our discussion:

Comparisons of pterosaurian pycnofibers and dinosaurian quill structures has led to the suggestion that the epidermal insulatory structures may be the primitive conditions for Ornithodira (i.e the most recent common ancestor of pterosaurs, dinosaurs and all their descendants; [135]). This hypothesis has been questioned by character optimization analyses [136], though they acknowledge that Early Jurassic theropod resting imprints with epidermal structures [137] makes proto-feathered Triassic theropods plausible. 

In the absence of definitive skin impressions, our model can add to the discussion on the need for insulatory structures in Coelophysis [138-140] despite the absence of skin impressions in basal theropods. We have not added such structures to Plateosaurus, as there are no prosauropod or sauropod body (or trace) fossils that demonstrate insulatory epidermal structures. 

We believe that this captures the spirit of what Reviewer 2 requested.

---

## [Decision Letter · Decision Letter 2]

13 May 2020

Modeling Dragons: Using linked mechanistic physiological and microclimate models to explore environmental, physiological, and morphological constraints on the early evolution of dinosaurs

PONE-D-19-26881R2

Dear Dr. Lovelace,

We are pleased to inform you that your manuscript has been judged scientifically suitable for publication and will be formally accepted for publication once it complies with all outstanding technical requirements.

With kind regards,

Ulrich Joger

Academic Editor

PLOS ONE

Additional Editor Comments (optional):

The last comment of Reviewer indicates that he has a different opinion than you on borrowing behavior. Yet you may leave your manuscript as it is for scientific discussion.

Reviewers' comments:

Reviewer's Responses to Questions

**Comments to the Author**

1. If the authors have adequately addressed your comments raised in a previous round of review and you feel that this manuscript is now acceptable for publication, you may indicate that here to bypass the “Comments to the Author” section, enter your conflict of interest statement in the “Confidential to Editor” section, and submit your "Accept" recommendation.

Reviewer #1: All comments have been addressed

Reviewer #2: All comments have been addressed

2. Is the manuscript technically sound, and do the data support the conclusions?

Reviewer #1: Yes

Reviewer #2: Yes

3. Has the statistical analysis been performed appropriately and rigorously? 

Reviewer #1: Yes

Reviewer #2: Yes

4. Have the authors made all data underlying the findings in their manuscript fully available?

Reviewer #1: Yes

Reviewer #2: Yes

5. Is the manuscript presented in an intelligible fashion and written in standard English?

Reviewer #1: Yes

Reviewer #2: Yes

6. Review Comments to the Author

Reviewer #1: Dear authors,

within the added texts you state that a burrowing behavior of juvenile Plateosaurus specimens cannot be excluded. I do not think that they have burrowed actively. The bauplan like the one seen in juvenile Prosauropods does not fit to burrowing behavior. Burrowers usually have short necks, strong claws, and if they are really specialised they show anatomical reductions like shortening of jaws, reducing of tooth number, reduction of the orbitae. None of these characters can be found in juvenile Plateosaurus specimens.

This is my only criticism concerning the actual manuscript.

Reviewer #2: During my most recent review of this manuscript I requested expanded discussion of some further context and caveats surrounding these results. I am pleased to say that I consider these points to have now been adequately addressed, and so am able to recommend publication in its current form.

7. PLOS authors have the option to publish the peer review history of their article (what does this mean?). If published, this will include your full peer review and any attached files.

Reviewer #1: Yes: Ralf Kosma

Reviewer #2: No

---

## [Editor Report · Acceptance letter]

14 May 2020

PONE-D-19-26881R2 

Modeling Dragons: Using linked mechanistic physiological and microclimate models to explore environmental, physiological, and morphological constraints on the early evolution of dinosaurs 

Dear Dr. Lovelace:

I am pleased to inform you that your manuscript has been deemed suitable for publication in PLOS ONE. Congratulations! Your manuscript is now with our production department. 

With kind regards,

on behalf of

Dr. Ulrich Joger 

Academic Editor

PLOS ONE